# ReefTEMPS: The Pacific Islands Coastal Temperature Network

Romain Le Gendre[1,6], David Varillon[2], Sylvie Fiat[1], Régis Hocdé[3], Antoine de Ramon N'Yeurt[4], Jérôme Aucan[5], Sophie Cravatte[6], Maxime Duphil[1], Alexandre Ganachaud[6], Baptiste Gaudron[2], Elodie Kestenare[6], Vetea Liao[7], Bernard Pelletier[8], Alexandre Peltier[9], Anne-Lou Schaefer[1], Thomas Trophime[7], Simon Van Wynsberge[10], Yves Dandonneau[8], Michel Allenbach[8] and Christophe Menkes[1]

[1]UMR 9220 ENTROPIE (Ifremer, IRD, Univ. Réunion, Univ. Nouvelle-Calédonie, CNRS), BP 32078, 98897 Nouméa CEDEX, New Caledonia
[2]UAR 191 IMAGO (IRD. Nouvelle-Calédonie), BPA5, 98948 Nouméa, New Caledonia
[3]MARBEC (Univ Montpellier, CNRS, Ifremer, IRD), Montpellier, France
[4]Pacific Centre for Environment and Sustainable Development (PaCE-SD), the University of the South Pacific, Suva, Fiji
[5]Pacific Community Centre for Ocean Science (PCCOS), 98848 Nouméa, New Caledonia
[6]Université de Toulouse, LEGOS (IRD/UT3/CNES/CNRS), Toulouse, France
[7]Direction des Ressources Marines, Papeete, French Polynesia
[8] Retired
[9]Météo-France, interregional office in New Caledonia and Wallis-and-Futuna, Noumea, New Caledonia
[10]Ifremer, IRD, ILM, UPF, UMR 241 SECOPOL, Vairao, 98725, Tahiti, French Polynesia.

*Correspondence to*: Christophe Menkes (christophe.menkes@ird.fr) and Romain Le Gendre (romain.le.gendre@ifremer.fr)

**Abstract.** While the rise in global ocean temperature continues its course, reaching 1.45+/- 0.12°C above pre-industrial level according to the World Meteorological Organization in 2023, marine heat waves frequencies and intensities increase. Consequently, coral reef ecosystems which are among the most vulnerable environments are strongly impacted with dystrophic events and corals experiencing increasing frequencies of bleaching events. That has devastating consequences for the Pacific Island Countries and Territories (PICTS) that strongly rely on these ecosystems. In-situ observation remains the best alternative for providing accurate characterization of long-term trends and extremes in these shallow environments. This paper presents the coastal temperature dataset of the ReefTEMPS monitoring network in which moored stations are implemented over a number of PICTS over a wide region in the Western and Central South Pacific from New Caledonia to French Polynesia. These in situ temperature time series are unique in several ways: in the length of some historical stations dating back to 1958 for the oldest, thus providing more than 65 years of daily data; in the number of countries sampled (16 PICTS) ; and in the variety of coral ecosystems monitored (from atolls to high islands and from barrier reef's external slopes to shallow and narrow lagoons). Measurement devices have evolved over the years to provide increasingly precise and frequent observations so that the ReefTEMPS network was endorsed as a French National Observation Service in 2020, a label ensuring quality controlled and open access data of long-term observations. All stations are publicly available in ASCII or formatted NetCDF files, either on the ReefTEMPS dedicated Information System which also allows quick visualisation of time series, or in the SEANOE marine data platform. All links and accesses to these temperature time series are provided herein. The longevity of these temperature time series allows diagnosing long-term trends, highlighting the influence of multiple processes on temperature dynamics (e.g., internal waves, cyclones, seasonal and climate modes) and documenting the time evolution of extreme events. All files are made publicly available in dedicated SEANOE repositories (DOI provided herein).

## 1 Introduction

Sea temperature is a key variable in oceanic, atmospheric and coupled ocean-atmosphere studies. It is an essential variable to be considered when characterising climate variability and climate change. In addition, it is also key for understanding marine ecosystems responses to thermal variability because of its wide influence on marine biogeochemistry and diversity (Kurylyk et Smith, 2023). It more particularly influences marine species spatial and temporal distributions (Pinsky et al., 2020; Righetti et al., 2019) and their life cycles (Dahlke et al., 2020). Understanding the evolution of oceanic temperatures is crucial to infer how global marine biodiversity and biomass will evolve as climate change is producing extremes that may not have been experienced by marine life before (Smale et al., 2019).

Since the 1980s, the advent of satellites has provided a better knowledge on how surface oceanic temperatures evolve at scales of ~ 25km (Minnett et al., 2019). Products such as OISST offer a retrospective view back to 1982 at 0.25° resolution (Reynolds et al., 2007). Lately, this synoptic capacity to observe surface temperature has strongly progressed into much higher spatial resolution with international efforts producing blended daily products up to ~1km resolution at global scale (e.g., MUR SST, Chin et al., 2017). This new higher resolution surface products have been complemented, since 1999, by in situ observations

of the water column temperature, with the launch of the global array of autonomous free-drifting profiling floats mainly in the open ocean (ARGO, Wong et al., 2020).

Yet, coastal and shallow water areas remain largely undersampled. First, Argo floats cannot drift in shallow waters, and at the coastal scale, even the highest resolution global satellite products are plagued by many sources of artefacts that cause remotely-borne temperature observations to strongly diverge from observed in situ estimates (Goebeler et al., 2022, Smit et al., 2013). Coastal areas often display high complexity and variability in terms of bathymetry, coastlines or freshwater inputs that create thermal micro-habitats that satellite data do not resolve properly. Resolution offered by satellites can also lead to a misrepresentation of true thermal extremes experienced at the coastal zone (Schlegel et al., 2017; Van Wynsberge et al., 2017). In addition, processes affecting infra-daily sea surface temperature variability (e.g diurnal heating signal, tidal signal or internal waves, Colin and Johnston, 2020) are invisible to most remotely-sensed techniques that only provide daily estimation of surface temperature. Some satellite measurements may provide these temporal scales (e.g Himawari, Kurihara, 2016) but over short time periods. Satellite products generally provide estimates of the upper 10-m temperature based on their radiometer measurements of the skin temperature and other parameters with inherent limitations to describe the water column or benthic thermal variability experienced by sessile organisms (Minnett et al., 2019).

At present, the only way to obtain true continuous temperature measurements in shallow water environments comes from moored observations. While those cannot assess the spatial scales that satellites cover, they provide ground truth temperature measurements of the water column at very high frequency and over long-time periods if moored observing systems are implemented in perennial manners. It is thus of crucial importance to maintain and enhance these arrays especially in small islands surrounded by coral reef environments where ecosystems goods and services are fundamental for people's well-being (Santavy et al., 2021).

Coastal observations are hence essential prerequisites to manage and mitigate risks, generate prediction of coastal hydrodynamics including temperature dynamics and create a continuous observing network from terrestrial to oceanic ecosystems (Malone et al., 2014). Knowledge about coastal sea water temperature variability is critical as it is part of the backbone of core biogeochemical and physical observations needed to inform management bodies and scientists on coastal events and processes (Bailey et al., 2019). In a warming world that exacerbates occurrence of extreme events such as marine heatwaves (IPCC, 2023), long term coastal monitoring of high-temporal-resolution-temperature is of crucial importance for making reliable assessment of these changes at all scales, from sub-diurnal to multidecadal (Goebeler et al., 2022; Salat et al., 2019). Shorter-term observations of temperature are also proving crucial for understanding mechanisms driving short-term temperature dynamics and for validating and setting up statistical or numerical modelling tools able to simulate thermal short-term variability (McCabe et al., 2010; Van Wynsberge et al., 2017). Misrepresentation of such short-term coastal processes may hamper our ability to perform long-term future projection for coastal ecosystems (Siedlecki et al., 2021)

Those general considerations on the need for in situ monitoring of temperature in coastal environments are particularly true for coral ecosystems. In these ecosystems, concerns about temperature effects have arisen since the 1998 global bleaching event. Although "localised" bleaching and dystrophic events have been reported since 1982 in the Pacific and Indian ocean as

well as in the Caribbean Sea (Goreau et al., 2000), the intensity and spatial extent of the 1998 event led to the awareness that global coral ecosystems may be durably endangered by climate variability (Hughes et al., 2017). This also stressed the necessity to better understand the complex relationships between coral bleaching and extreme ocean temperatures. In the tropical Pacific, the health of coral reef ecosystems is a fundamental issue as it has a major impact on food security as well as sources of income for Pacific islanders (Bell et al., 2017). As ocean warming and heatwaves are actually recognized as the most significant and growing threats to coral reefs (IPCC, 2023), in situ temperature monitoring appears of fundamental importance to better assess their fate in the future by being able to document lethal thresholds from in situ data and/or possibly find more heat-tolerant coral reef populations (De Carlo et al., 2019; Rivera et al., 2022) for example.

Temperature variability within coral reef ecosystems (such as lagoons, outer reef slopes, reef flats or terraces) can be controlled by a variety of physical drivers of both oceanic and atmospheric origins (Herdman et al., 2015; Grimaldi et al., 2023). Moreover, interactions of physical processes (tides, wind, waves down to turbulence within coral canopy) with complex bathymetry induced by coral reefs geomorphology can lead to thermal microclimates (Reid et al., 2020). The resulting local thermal signatures can thus be observed only by the means of in situ monitoring and strongly supports field observations for understanding coral bleaching (Safaie et al., 2018; Green et al., 2019) or coral cover spatial heterogeneity (Rogers et al., 2016). Toward that end, several in situ coastal water temperature monitoring strategies have been launched since early 2000s in the tropical Pacific, either at a regional scale (e.g Potemra et al., 2017 for the Pacific Island Ocean Observing System : PacIOOS), or at country scales (e.g Palau – Coral Reef Resarch Foundation et Colin, 2018; Australia – Lynch et al., 2014; Federated States of Micronesia, Pohnpei – Rowley et al., 2019; French Polynesia, Morea LTER Network – Leichter et al., 2013).

Along these lines, the ReefTEMPS initiative has been federating past and on-going coastal scale projects or temperature datasets in the South-Central and South-West Pacific islands. One of the strengths of this network is to maintain long-time observational efforts for quality measurements so that it gathers a number of in situ coastal temperature data dating back from 1958. The ReefTEMPS monitoring initiative is thus dedicated to documenting a range of temporal scales from long-term trends of coastal ocean temperature associated with climate change and their impacts on coral reef systems to shorter time scale processes shaping coastal thermal regimes within these ecosystems. In addition to honouring the observational effort, the origins and scientific values of the past gathered datasets from different institutions, this paper aims to present the philosophy and quality of this coastal reef monitoring current network and its future directions, in order to ensure the continuity of such crucial observations. This paper is also a means by which to advocate future and more global collaborations on these observations that will ensure the sustainability of the network regardless of the turmoils linked to funding uncertainties.

The paper is organised as follows. After a description of the history and current status of ReefTEMPS in section 2, section 3 provides details on sampling devices used since the beginning of observations. Section 4 sets out the overall strategy and methods that ensure data quality while part 5 presents the philosophy of data management and dissemination. Finally, after a brief presentation of some key applications of such temperature data in section 6, section 7 is dedicated to the perspectives and future evolutions of ReefTEMPS.

## 2 ReefTEMPS: Coastal temperature monitoring in Pacific Islands

### 2.1 History

The ReefTEMPS (Pacific Islands Coastal Temperature Network) initiative, was officially launched in 2010 by the GOPS (Grand Observatoire de l'environnement et de la biodiversité terrestre et marine du Pacifique Sud), by federating existing coastal monitoring strategies and datasets and adding numerous sites of measurements in the South Pacific. In practice, the adventure actually began much earlier. As early as 1958, in Nouméa (New Caledonia, NC), ORSTOM (Office de la Recherche Scientifique et Technique Outre Mer, now IRD, Institute of Research for Sustainable Development)'s oceanographers were convinced of the crucial value of repeated and prolonged measurements of sea parameters (temperature and salinity). Using the material resources available at that time (oceanographic bucket), they worked hard to maintain daily observations of temperature and salinity at the first long-term lagoon monitoring station of Anse Vata– Nouméa (Dandonneau, 1986, Fig. 1 - Appendix B1). Ten years later, in 1967, a second historical station was set up, closer to the open ocean, on the islet of the Amédée lighthouse (Fig. 1 - Appendix B1). The foundation of the ReefTEMPS network was born.

From 1992 to 2009, management and continuity of the existing monitoring network in New Caledonia lagoons has been steered by IRD with the support of the Zoneco program (https://www.zoneco.nc/) with the start of new observation stations around the mainland of NC, on both the west and east coasts and both northern and southern lagoons. This geographical extension began mainly in 1997 when electronic sensors replaced manual sampling. 2010 was the official birth year of the ReefTEMPS framework driven by the GOPS. In addition to major improvements on data archiving and dissemination infrastructures (Hocdé & Fiat, 2013), ReefTEMPS expanded to other PICTS during 2011-2015. In 2011, with financial support from the Australian Agency for International Development (AusAID), the Pacific Community (SPC) launched a project to help Pacific Island countries in setting up pilot projects to monitor coastal fisheries and associated habitats. In this context, a dozen sensors were deployed in Marshall Islands, Cook Islands, Papua New Guinea, Micronesia, Tuvalu and Kiribati and were integrated in ReefTEMPS. In 2012, through a collaborative initiative, management of the historic stations on Wallis and Futuna was entrusted to the University of New Caledonia. The same year, the Pacific Centre for Environment and Sustainable Development (PaCE-SD) at the University of South Pacific in Fiji joined the ReefTEMPS initiative and began observations in Fijian coastal waters, thus developing a long-lasting collaboration with ReefTEMPS which endures to this day. Finally, in 2021, the Direction des Ressources Marines de Polynésie Française (DRM) also integrated ReefTEMPS by including their historical data from the French Polynesian lagoon network RESOLAG (Liao et al., 2023) to the ReefTEMPS dataset and has since become another major partner of the network.

As an international observation network based in both the French Pacific territories and the Pacific Island states (Hocdé et al., 2021), ReefTEMPS has been a key asset in the creation and design of France's multi-agency Research Infrastructure for coastal ocean observation ILICO (Cocquempot et al., 2019). Since 2019, ReefTEMPS has been one of the nine National Observation Services (SNO) integrated in ILICO. These networks are accredited through a peer-reviewed evaluation process overseen by French national research agencies every 5 years. ReefTEMPS was labelled as SNO by the french governmental "Ocean-

Atmosphere" commission for the 2020-2024 period and for the three parameters: temperature, conductivity and pressure. As
a labelled network, ReefTEMPS is required to acquire and disseminate openly data of international quality standards.

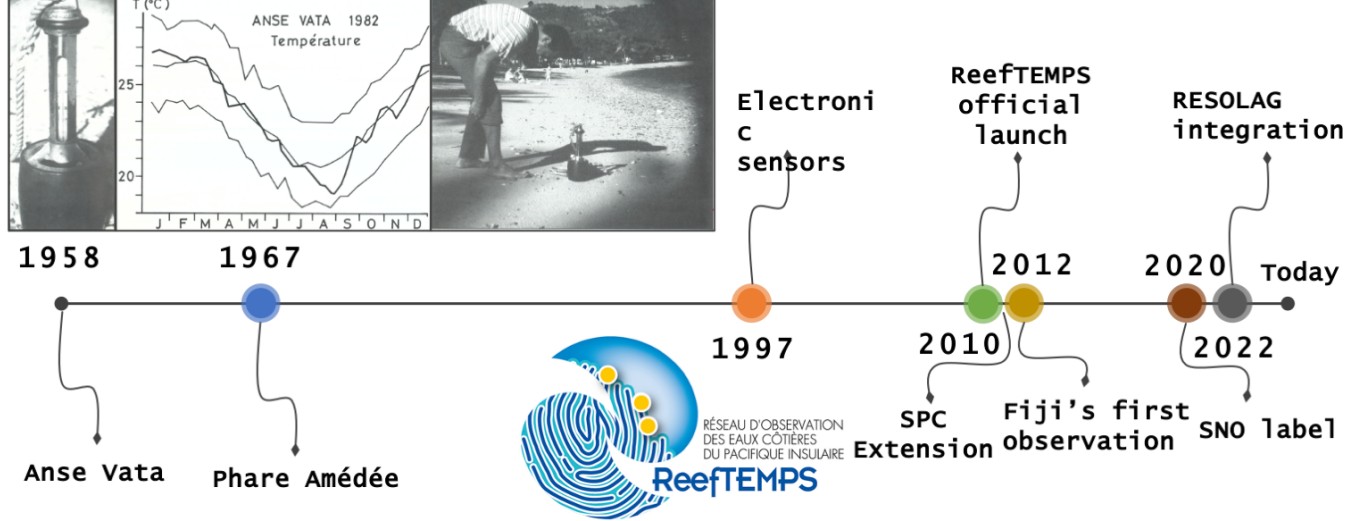

**Figure 1: Timeline of the main events of the ReefTEMPS Network. During the first period until 1997, bucket measurements were**
**done as depicted in the inserted photos from Dandonneau (1986). Left panel of the insert: zoom on an oceanographic bucket. Centre:**
**seawater temperature at Anse Vata station using bucket (bold line: 1982 time series, lights lines represent average, minimum and**
**maximum through the year from 1958 to 1982 observations). Right: scientist reading temperature value on an oceanographic bucket.**
**2.2 The current ReefTEMPS Network**
The tropical and subtropical Pacific is the area of the world oceans that supports the largest habitat for coral reefs and is home
to the greatest coral species richness (Maragos and Williams, 2011; Fig. 2 upper right panel). The ReefTEMPS temperature
monitoring network encompasses the three regions of Oceania (Micronesia, Melanesia and Polynesia), covering 16 PICTS
(see Fig. 2, Tables A1 & A4, Appendix B2) and extending roughly from 10 to 30°S and from 134°E (Palau) to 134°W (Gambier
islands). Such huge spatial coverage is a challenge to maintain over time and some stations have been discontinued due to
fluctuating collaborations and fundings (57 stations interrupted). The duration of time series ranges from 6-8 months for the
shortest series to more than 65 years for the longest (Anse Vata station, New Caledonia). 26 stations have more than 10 years
of observations (approx. 20% of the monitored sites). The total observation time, covering periods between the start and end
dates of all stations sums actually to 320744 days, equivalent to approximately 878 years of data. The study sites that have the

higher numbers of monitoring stations and currently contribute the most to the observations of coastal temperature are New Caledonia, Fiji and French Polynesia, which constitute the secure and core observations of ReefTEMPS. New Caledonia (Fig. 2 - Bottom left), due to its history of coastal observations, represents the "backbone" of this network with both the largest number of monitored sites (53) and the longest time series. Most stations are located in its southwest lagoon but some long-term sites are also spread out further north and on the east coast of the mainland ("Grande Terre"), as well as on remote reefs (e.g Entrecasteaux reefs, Chesterfield islands). Fiji currently has 15 monitored sites around Viti Levu Island, Beqa Island, the Vatu-i-Ra Passage, the Lau Group and the northernmost island of Rotuma. In French Polynesia ReefTEMPS covers the 5 main archipelagos, sampling both atolls and high islands lagoons with a total of 20 stations.

Overall, to date, the ReefTEMPS network currently comprises 118 monitoring temperature stations (61 currently active) with mean duration of observations above 2430 days. Since time series are generated by instrument type, this corresponds to a total of 132 files. In terms of depth, sensors are distributed between 0.5m and 60m (61 % in the 0-10m, 33% in the 10-20m, and 6% > 20m) (see Appendix A - Table A4). The vast majority of observation stations consist of measurements at a single depth, while two stations in New Caledonia (Uitoe and Hienghène) are equipped with multiple vertical levels of instrumentation.

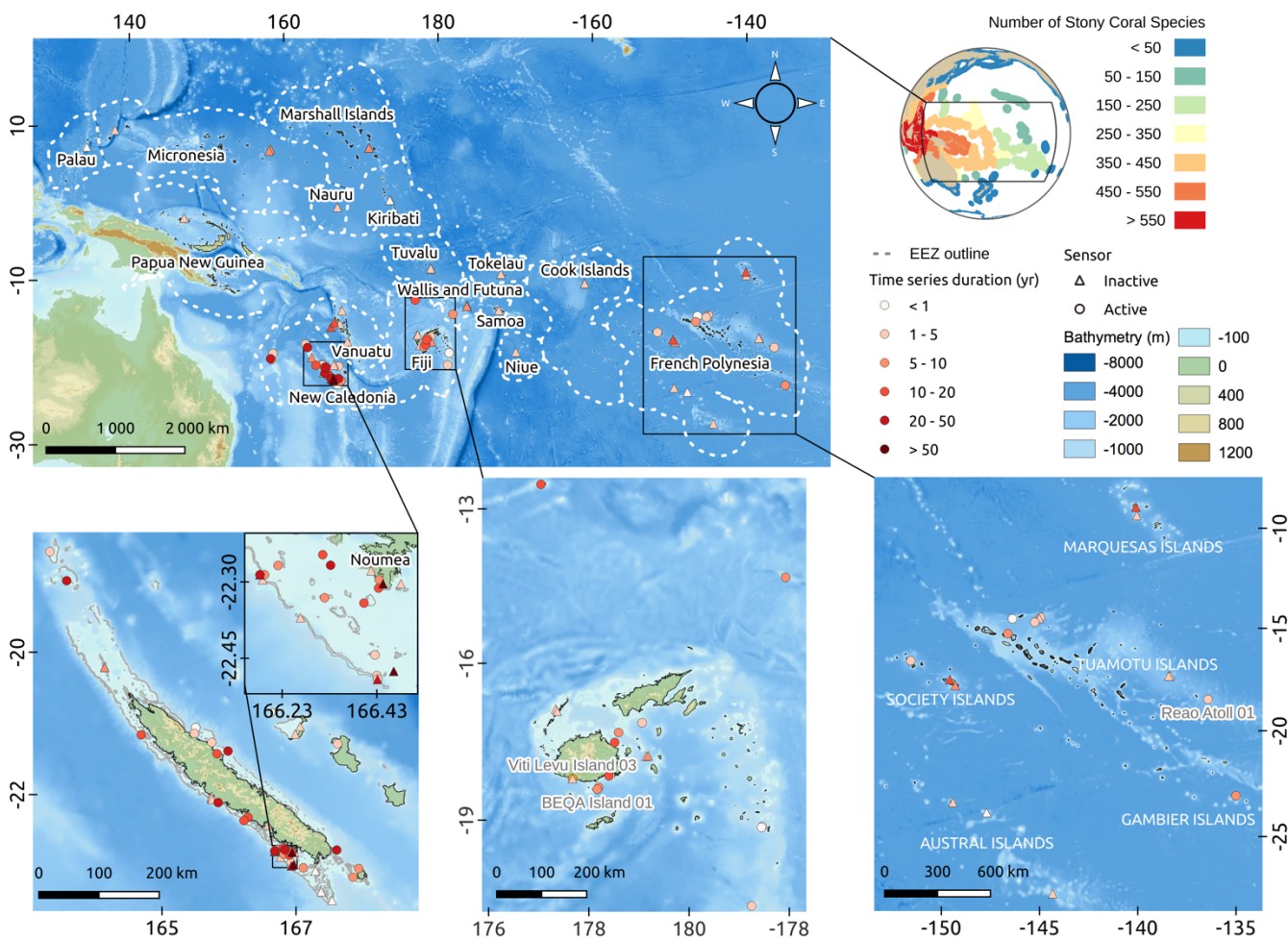

**Figure 2: Overview of geographical distribution and length of time series (in years) of ReefTEMPS monitoring stations. Detailed zooms are provided for New Caledonia, Fiji and French Polynesia. Circles/triangles indicate respectively active/inactive stations. The upper right panels depict the number of stony coral species across the world (from The Atlas of Global Conservation, Hoekstra et al., 2010) illustrating the coral reef context in which ReefTEMPS is set. Bathymetric data used come from GEBCO grid (GEBCO Compilation Group, 2022).**

## 3 Sampling devices

Due to the wide temporal range of the ReefTEMPS dataset, measurement methods have evolved over the years in line with technological advances. Starting from simple observations with an oceanographic bucket deployed from the shore by a human operator (see Fig. 1), the network has grown to include a variety of automatic sensors with increasing accuracy, frequency of acquisition and capacity of storage. Most of the instruments used now are autonomous compact loggers containing internal batteries and memories, deployed by scuba diving and fixed on the seabed (see Fig. 3). Moorings have been designed to be adapted to the habitats and to withstand heavy agitation such as the ones induced by cyclones or storms. To prevent sensors

from biofouling, mechanical damage or wildlife, they are all deployed inside plastic cylinders with holes that allow water
circulation. A few sites (especially in French Polynesia, see buoys section) were also initially instrumented using buoys but
this sampling strategy is now replaced by moored sensors to be congruent with the whole network.

**3.1 Oceanographic bucket**

For the two long-term sites of New Caledonia, Anse Vata and the Amédée islet, data were first collected using the
oceanographic bucket (see Fig. 1). This device was as simple and robust as a water-taking bucket equipped with a thermometer
and deployed using a rope to collect water. It allowed temperature measurements to be taken with an accuracy close to 0.1°C
and had been used daily for nearly 47 years. The nominal acquisition time for both stations was 7am local time and the targeted
depth using the bucket was ~0.5m.That method was abandoned in 2005 to move to more automatic measurements. At the
Amédée station, the construction in 1977 and extension in 1993 of a pontoon slightly shifted the sampling point from the initial
position from the beach, moving it away from the shoreline by 44m, then 64m. That changed to 4.5m with the arrival of
autonomous electronic loggers. In French Polynesia, two stations had also been sampled daily using buckets in Tahiti (Society
Islands, from 1979 to 1989) and in Ua Pou (Marquesas Islands, from 1986 to 1989).

**3.2 Compact autonomous loggers**

From 1997 to 2009, a few main initial sensor brands were used for monitoring coastal temperatures in New Caledonia, French
Polynesia and Wallis Island. The first set of electronic and autonomous sensors deployed were HOBO®, for which various
models were successively used (Stowaway : https://www.onsetcomp.com/resources/documentation/1513-stowaway-xti ;
Optic Stowaway : https://www.onsetcomp.com/resources/documentation/1086-k-man-optt ; UTBI-001 TidBit :
https://www.onsetcomp.com/products/data-loggers/utbi-001; last access: 5 September 2024). Depending on the brand, the
accuracy ranged from 0.2 to 0.4°C, but these sensors provided a higher temporal resolution compared to the punctual
observation using a bucket. They provided infra-daily resolution, acquiring data continuously at frequencies between 10 and
30 min. Autonomous loggers from RBR Ltd, the RBR TD1060 were also initially deployed in New Caledonia. In addition to
temperature (accuracy 0.002°C, drift ~0.002°C/year; manufacturer's manual), they also provided observations of pressure.
Due to several logger failures or drifts, these RBR sensors were gradually abandoned. At last, the Uitoe station (external slope
of the barrier reef, west of New Caledonia) was equipped since 1992 with a Seacat SBE16 from SEA-BIRD Electronics Inc.,
which samples temperature (accuracy 0.01°C, resolution 0.001°C) but also conductivity.
With the birth of ReefTEMPS in 2010 and its associated requirements, as well as the technological developments that occurred
in oceanographic instrumentation, the compact loggers fleet has evolved towards models with longer autonomy and greater
accuracy while measuring additional parameters. The GOPS has led a major effort to rejuvenate and homogenize the
instrumental fleet. Depending on monitoring sites and scientific objectives (e.g additional observations of level and salinity),
the choice fell on a new generation of robust devices that allows long-term deployments (from 6 month up to 2 years) with
minimum battery costs while being strongly reliable. Since 2010, SBE56 temperature sensors were moored (SEA-BIRD
Electronics Inc.; https://www.seabird.com/sbe-56-temperature-sensor/product?id=54627897760, last access: 5 September
2024). These SBE56 loggers allow recording fast (1min sampling rate), highly accurate temperature measurements (accuracy
of 0.002°C, +- 0.002°C drift/year), and provide enough battery and storage autonomy to remain deployed underwater for up
to 2 years. For monitoring stations where water level dynamics is of interest, the sensors used in ReefTEMPS are now two
models from RBR Ltd. namely, RBRduo T.D and RBRduet T.D (https: //rbr-global.com/, last access: 5 September 2024).
These RBR loggers are used to record not only temperature (initial accuracy of 0.002°C, +- 0.002°C drift/year) but also
pressure that provides information about sea-level dynamics. Finally, on stations impacted by massive freshwater inflows,
temperature is monitored using the Infinity-ACTW loggers from JFE Advantech Co., Ltd. (https://www.jfe-
advantech.co.jp/eng/assets/img/products/ocean-infinity/INFINITY-CTW(E)_201704.pdf, last access: 5 September 2024)
which reliably samples temperature (accuracy +- 0.01°C, resolution 0.001°C) and conductivity (salinity).

**3.3 Multi-parameter buoys**

In French Polynesia, RESOLAG, a program dedicated to the long-term monitoring of pearl farming atolls, started in 2018
(Liao et al., 2023). The aims of the deployed sampling strategy were initially double: first to acquire multiple parameters
(temperature, salinity, fluorescence, turbidity, dissolved oxygen) to understand the link between environment variability and
performance of pearl farming activities (e.g spat collecting, pearl quality). The second objective was to provide pearl farmers
and stakeholders with a real-time view of the lagoon's state, particularly temperature data, to make their spat collection seasons
more efficient by improving their understanding of precise interseasonal periods. For this purpose two kinds of real-time multi-
parameters buoys by NKE (Smatch and Sambat models; https://nke-instrumentation.com/; last access: 16 May 2024) were
deployed in 7 different lagoons, sampling parameters around 3m at 1 hour frequency. Concerning the thermistors, the
manufacturer's manual for the thermistors indicates for both buoys an accuracy of 0.05°C and a maximal resolution of 0.003°C.
In 2023, due to some problems with live transmission and sensor maintenance, the RESOLAG strategy shifted to the use of
moored loggers and, to be consistent with the ReefTEMPS logger strategy, choice fell on SBE56 and RBR Duet instrument
(see section 3.2 above).

**4. Processing and quality control**

**4.1 Overall strategy**

Figure 3 presents the global data life cycle of the ReefTEMPS temperature time series. Data processing and quality control
have been conducted in a standardised manner since 2010 to ensure both consistency of observed time series and diffusion
using international oceanic data standards. Since 2010, maintenance and recalibration of instruments have been conducted at
recommended intervals by or in accordance with the manufacturers (every 5 years for Seabird loggers, every 2-3 years for JFE
Advantech and RBR loggers) to ensure reliability and quality of values observed. Recently, an intercomparison procedure for
sensors compared with a reference SBE56 sensor was implemented to ensure that the sensors do not differ by more than
0.005°C from the reference sensor. Prior to 2010, the reliability and accuracy of the devices used were lower (see Section 3)
and maintenance frequency was not really fixed and fluctuated according to available funds.
Early 2025, a major overhaul of the entire database, quality flags and processing states, has been carried out. Each temperature
measurement of each time series now has an associated quality code (QC, see Table A5). Based on the knowledge on sensors
and sampling devices accuracy, quality flags have been attributed to each measurement according to instrument type and
family. Globally, buckets, Onset and NKE sensors have been flagged to "Probably good data" (this mainly concerns data prior
to 2010). RBR and Seabird sensors have been flagged to good data since, in our opinion and with our expertise, they provide
more reliable temperature data. A python graphical tool has been used for inspecting all temperature time series and modifying
quality flags. This tool lets you zoom in to perform visual check of all time steps, display satellite temperatures (e.g OISST,
OSTIA), perform basic statistical tasks (e.g remove eventual duplicate data, tests values over/below threshold) and finally
assign a different  quality code to desired measurement. This qualification stage will now be fully integrated in the data life
cycle (see Figure 3) for future data integration in the temperature database.
Finally, a dedicated nomenclature for files based on international standards (either raw or processed) was also implemented
(see tables A2, A3, A4 for information on stations, instrument types and processing states and Fiat et al. 2024). Dataset file
names read as follows: ConventionFormat_CodeSite_Startime_ParameterType_ProcessingState_InstrumentType_Depth. For
example, filename 'OS_POINDI01_199710_TEMP_2B_TR_125.nc' indicates that this time series is formatted following
OceanSITES conventions (OS ; https://repository.oceanbestpractices.org/handle/11329/874.2; last access: 5 September
2024), taken at POINDI01 monitoring station (Poindimié station on NC east coast), beginning in October 1997, processed up
to "Quality controlled data" (2B) processing state (See Appendix A Table 3), with instruments belonging to Thermistor class
(See Appendix A Table 2), moored at a 12.5 meter-depth and provided in NetCDF (.nc). To avoid decimal numbers in the
filenames we have chosen to indicate depths in decimeters. The global data life cycle (including processing and quality
steps) is described hereafter and depicted in Figure 3 diagram. The files generated at each stage are stored on secure drives.
1. Instruments are replaced (or moored if this is a first deployment for a new station) by scuba diving at frequencies that
depend on their characteristics (from 3 months up to 2 years in water). Each replacement is referred to in the database
as a "measurement cycle".
2. Upon replacement raw files are retrieved using dedicated manufacturer's software and first converted in ASCII format
and named following nomenclature rules.
3. Time series from each measurement cycle are then carefully visually inspected using specific softwares (ferret or
matlab routines) to ensure removal of obviously "bad data" (e.g out of water observations), converted into NetCDF
cycles, and checked to ensure measurement cycles are properly connected. At this stage the corresponding processing
state of the time series remains level 1A (see Appendix A3).

4. Measurement cycles are then imported into the DB Oceano database (PostgreSQL)

5. Datasets from each retrieval are then exported into NetCDF OceanSITES format (https://repository.oceanbestpractices.org/handle/11329/874.2; last access: 5 September 2024), with metadata following the Climate and Forecast metadata conventions (CF : https://cfconventions.org/; last access: 5 September 2024), and finally.

6. Using a python graphical tool, a qualification check (see above) is performed on the new cycle of observations and if necessary, modifications are performed on corresponding QC in the database. This qualification check, performed by a scientific expert in tropical coastal temperature dynamics, enables re-updating database and affecting processing state 2B to these temperature time series (see Appendix A3).

7. Fully processed NetCDF files are then exported into the ReefTEMPS Information System (IS) which allows delivering datasets in different formats and/or using different web services based on specific and standardised protocols (see 5.1).

On April 15, 2025, the global archive of temperature time series contains 132 quality-controlled temperature files: 125 files are at processing states level 2B and 7 files at 3B. The seven 3B files come from Temperature/Pressure sensors deployed at very high frequency (1Hz or 2Hz) to compute wave parameters. Temperature for these stations is therefore resampled to 30 minutes.

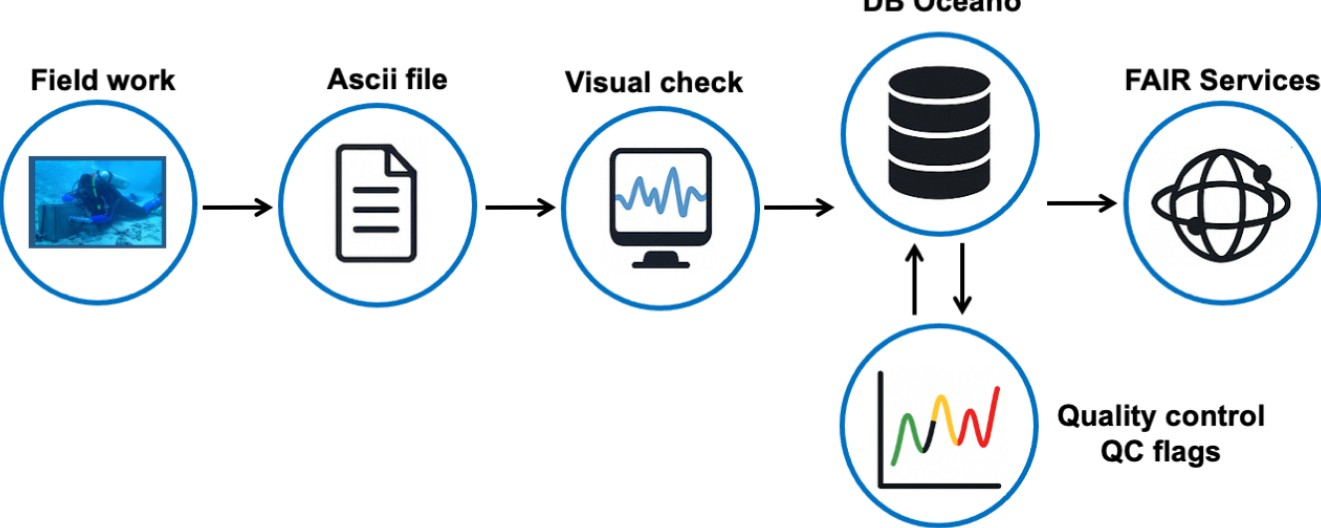

**Figure 3: Data life cycle of ReefTEMPS temperature time series. Photo: Batiki island sensor installation (Credit: Partners in Community Development, Fiji).**

**4.2 Long-term monthly homogenised files**

The instrument precision and targets of ReefTEMPS have evolved over time, starting with studies of daily-to-seasonal variability, then moving to longer term variability. Observations acquired before 2010 using oceanographic bucket or Hobo sensors suffered from a lack of precision or potential drifts. However, studies of the effects of climate change on coastal temperature require access to long homogeneous time series with sufficient precision as temperature trends detected since 1950 globally do not exceed a few tenth of degrees/decade (Cavarero et al., 2012; IPCC, 2023). Hence to avoid misinterpretation in long-term trends due to sensor turnover, displacement or change in the sensor environment, an homogenization procedure was applied to the two historical time series at Anse Vata and Phare Amédée stations in New Caledonia. That allowed providing daily homogenised time series for the longest records with which to look at climate trends. The procedure applied for building homogenised monthly long-term time series is described in depth in Guyennon (2010). During the first decades of observations (1958-1997), measurements using buckets targeted a sampling at 7 a.m. local time every day although some measurements were taken between 5 and 10 a.m. Depending on the month of the year, this sampling time difference can lead to temperature differences of up to 0.4°C. Thus, the first part of the procedure was devoted to readjust these data to be consistent. For that purpose, the HOBO sensor period (1998-2010) was used for each station to compute average daily temperature variations for each month and then perform adjustment of bucket data to represent only the 7 a.m. temperature regardless of sampling time. The second step of the homogenization procedure aimed to correct bucket data to be representative of the daily mean for each day. Common measurement periods between sensors and buckets (80 months for Anse Vata and approx 30 months for Phare Amédée) were used to quantify, for each month, the differences between bucket values and daily sensor averages. These differences were then applied to the bucket period to provide data series representative of the daily mean temperatures. Monthly mean temperature time series were computed for each station. Finally, detection and correction of artificial shifts were performed using the PRODIGE software from Météo-France (theoretical basis presented in Caussinus et Mestre, 2004) for the 1958-2010 period. After 2010, SBE56 sensor data (deemed much more accurate) were averaged monthly and concatenated to finally obtain two monthly long-term series for Anse Vata (1958 - 2023) and Phare Amédée (1967-2023). Homogeneity assessment tests were carried out using RHTest V4 (Wang et al, 2010) and revealed no significant breakpoints. The figure B4 in appendix B displays the monthly homogenized data versus raw ones.

**5. Data management and dissemination - Open access**

Prior to ReefTEMPS, the data was centralised on a database referred to as "DB-Oceano" (PostgresSQL database management system), which was developed by IRD in the early 2000s for managing data from marine sensors. The database framework was inspired by the one initially built by the multi-partners Coriolis Project (https://www.coriolis.eu.org/). The first version of the ReefTEMPS Information System (IS) was then put into production in 2011-2012 (Hocdé & Fiat, 2013). Then, several updates of the information system took into account technological changes and offered new functionalities to both data

managers and users (Brissebrat et al., 2017). Now the ReefTEMPS IS uses DB-Oceano with a workflow manager (Apache Airflow, implemented in 2023) around which web servers are deployed to distribute/share data. The infrastructure is designed around the concept of micro-services and is fully containerized using docker technology, ensuring good system portability and the possibility of upgrading to distributed servers for better load balancing. The workflow manager automates the integration of new data by establishing a set of management rules according to the results of previous tasks (Appendix B Fig B.3).

Overall, the architecture of the ReefTEMPS IS is designed to ensure data longevity, optimise accessibility, enable widespread dissemination and ensure interoperability with other systems (Fiat, 2015, Fiat et al., 2021). These concepts are in line with the FAIR principles: Findable, Accessible, Interoperable and Reusable (Wilkinson et al., 2016). The ReefTEMPS database is provided as an open resource under a Creative Commons Attribution-ShareAlike 4.0 International license (CC BY-SA). The core of the datasets diffusion engine used on the website (https://www.reeftemps.science/) consists of interactive map showing the location of monitoring stations via Web Map Service (WMS-OGC) and Web Feature Service (WFS-OGC) geographic services. Once a station has been selected by the user, datasets can be downloaded in multiple formats (NetCDF using OceanSITES format, Ascii file, or Comma Separated Value files) via different sharing protocols/servers (Thredds server and OpenDAP protocol, Sensor Observation Service (SOS-OGC)). A dedicated visualisation service is also available to explore time series on the website, using ad hoc python web routines. Finally, the whole ReefTEMPS data archive is also accessible through Digital Object Identifiers (Varillon et al., 2025: DOI:10.17882/55128 and Liao et al., 2025: DOI:10.17882/82291) and is updated every six months on the Seanoe data repository. Each release of the semestrial whole dataset is identified by a specific and additional key (i.e https://doi.org/10.17882/55128#107183 for the 2024-01 release, https://doi.org/10.17882/55128#103428 for the 2023-07 release, etc). Nevertheless, the ReefTEMPS archive DOI is unique and common to all archive releases, which allows it to better track data usage statistics. The ReefTEMPS Data Management Plan describes the life cycle of ReefTEMPS data from their acquisition to their dissemination, including the steps of processing, archiving, etc. (Ilico, 2023). Figure 4 presents an overview of the data portal page and the associated services.

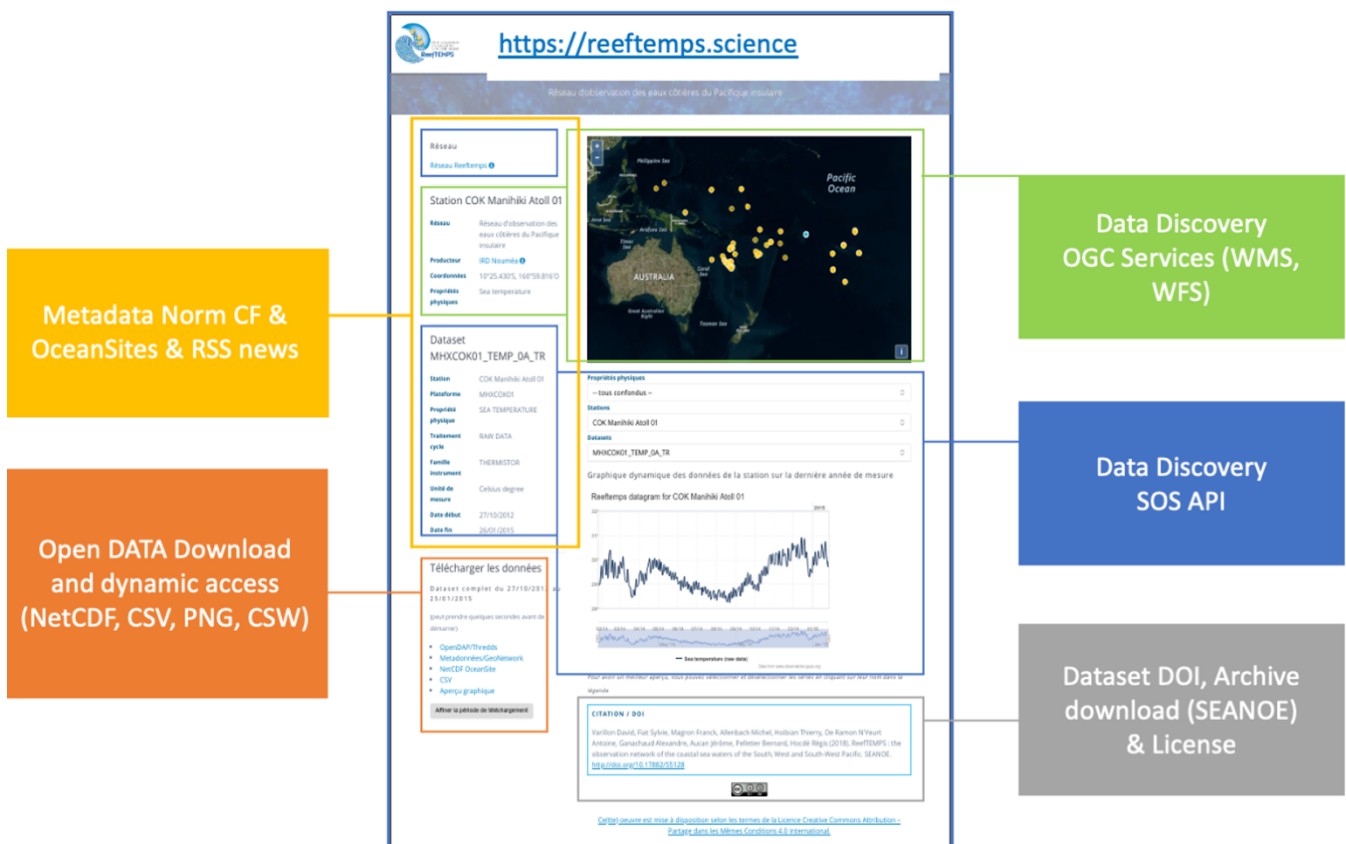


**Figure 4: Data access portal page and associated services**


## 6. Some examples of key applications


### 6.1 Capture and document extreme events


With the increasing frequency, intensity and duration of Marine Heatwaves (Oliver et al., 2018), in situ temperature
observations are crucial for understanding the impact of true thermal variability on coral ecosystems. Figure 5 shows extracts
from 3 chosen time series during austral summers 2016 (for Fiji and New Caledonia) and 2024 (for French Polynesia) where
elevated temperature negatively impacted the health of ecosystems and wildlife (Holbrook et al., 2022; Dutheil et al., 2024).
For the sake of illustrating the benefits of in situ observations, widely used daily L4 SST products are also displayed on each
subplots for the nearest points to the ReefTEMPS stations. The two selected products are respectively OISST V2
(https://psl.noaa.gov/data/gridded/data.noaa.oisst.v2.highres.html ; last access: 5 September 2024) depicting SST at 1/4° and
OSTIA SST (https://data.marine.copernicus.eu/product/SST_GLO_SST_L4_REP_OBSERVATIONS_010_011/description ;
last access: 5 September 2024) at 0.05° resolution. First, Viti Levu 03 station in Fiji, moored at 12m depth on the oceanic side


of the Votua lagoon, showed a sharp increase in temperature from January 15, peaking at nearly 31.25 °C on Monday 8[th] February 2016. On the same day, thousands of dead fish and invertebrates were found on the beaches near the village of Votua (Holbrook et al., 2022). Then category 5 tropical cyclone WINSTON re-entering the area on February 20[th], induced a strong cooling by more than 5°C, participating in the demise of that massive marine heat wave (Dutheil et al., 2024). At the same time the Anse Vata station in New Caledonia, located more than 1250 km from Viti Levu and moored inside the south-west lagoon at 2m depth, showed the same tendencies of rising temperature prior to March 2016. There, temperatures began to rise rapidly from mid-January onwards and also peaked at 30.7 °C on Monday 8[th] February 2016. Daily maximum temperatures exceeded 30°C for about twenty days, which is between 2.5 and 3°C above the climatology computed for the 1997-2023 using Hobday et al. (2016) (see Figure 5.b). It had strong consequences on corals: the first documented massive coral bleaching event in New Caledonia's lagoons occurred during that February 2016, while that lagoon had been relatively unscathed until then (Payri et al., 2018). The third major event illustrated here occurred in 2024 in Reao atoll lagoon (orange line) in French Polynesia where the important population of giant clams (*Tridacna maxima*) provides significant incomes and food to inhabitants through fishing and aquaculture practices (IUCN, 2021). In 2024, daily maximum temperatures frequently reached or exceeded 31.5°C for about a month from the end of February onwards (even reaching max values of 31.8°C at the end of March), and always remained above 29.9°C during 40 consecutive days. The consequences of these prolonged high temperatures highly affected giant clams, with 57% of exploitable giant clams totally bleached, and 43% partially bleached, as estimated on April 1[st] 2024 in the area around the location of this thermistor.

These three iconic examples associated with heat waves demonstrate the crucial importance of such in situ observations for a better understanding of thermal tolerance, physiological damages and resilience of tropical marine organisms towards heat stress. Indeed, while satellites tend to capture roughly the same low frequency temperature dynamics, large biases (more than 2°C) are present and may prevent the study of ecosystem vulnerability. Moreover, these time series (Fiji versus New Caledonia) also illustrate the potential of such a geographically extensive network for studying spatial variability of coastal temperatures across regions which can be very useful to study the regional heterogeneity of coastal thermal responses to climatic modes such as ENSO. Finally, at local scales, a high density of sensors inside the same lagoon for example can also provide valuable information for understanding smaller scale spatial variability which are not captured by state-of-art current satellite measurements such as MUR (Van Wynsberge et al., 2020).

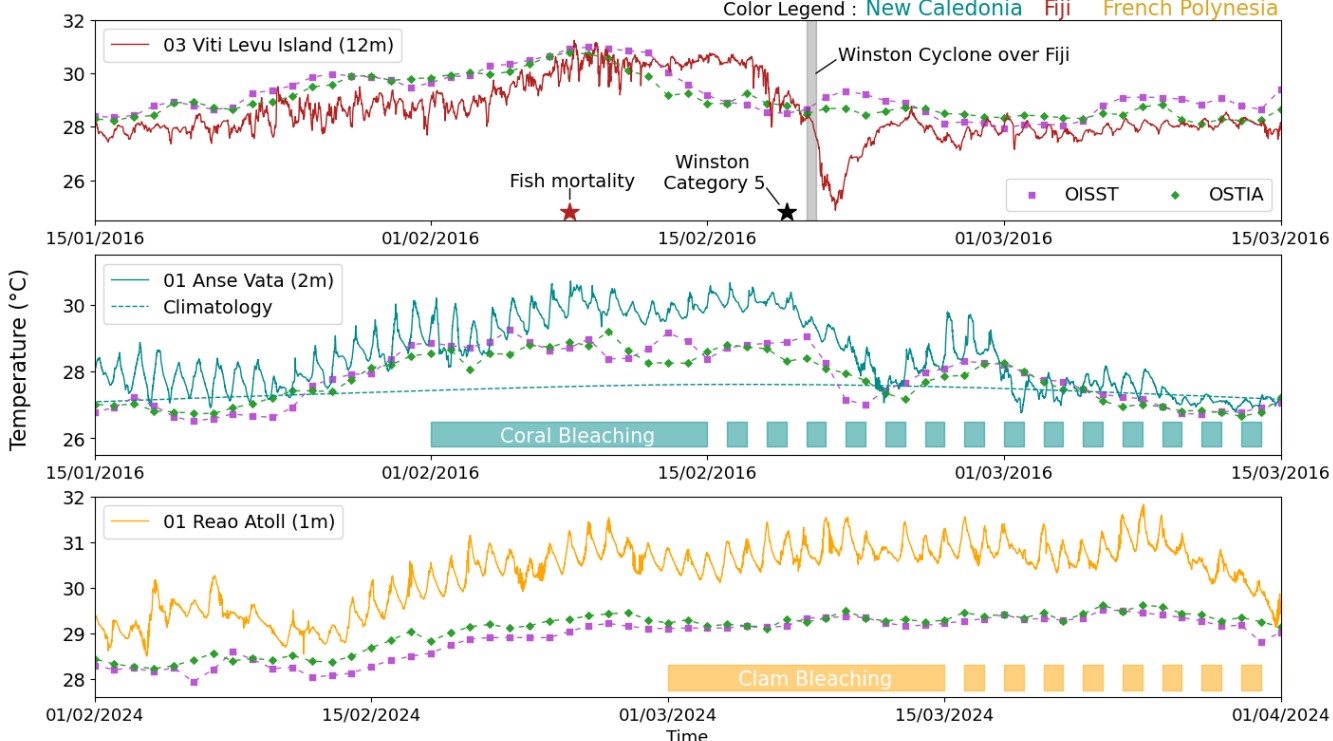

**Figure 5 – a. Temperature time series during austral summer 2016 at Viti Levu Island 03 station (Fiji, 12m, red line) b. Temperature**
**time series during austral summer 2016 at Anse Vata station (New Caledonia, depth 2m, dark cyan line) c. Temperature time series**
**at Reao 01 station (Reao atoll lagoon, French Polynesia, depth 1m, orange line). Daily Sea Surface Temperature from Satellite**
**products are plotted in purple for OISST V2 and green for OSTIA. Dates of the triggered ecosystem impacts are displayed on each**
**subplot (red star for fish mortality in Fiji, dark cyan bar for coral bleaching in New Caledonia and orange bar for clam bleaching**
**in French Polynesia). Dotted lines indicate that impacts on coral and clams have continued over time (with no precise end date to**
**give).**
**6.2 Characterise physical processes at various timescales**
The temperature records from the ReefTEMPS network demonstrate the importance of capturing physical processes operating
across multiple temporal scales. These measurements enable the differentiation of high-frequency variability, such as tidal or
diurnal fluctuations, from lower frequency signals associated with seasonal or interannual dynamics, thereby providing a
comprehensive understanding of coastal oceanographic processes. Figure 6 shows examples of physical processes affecting
temperature at different timescales as captured by the ReefTEMPS network. Here again, to highlight the crucial importance of
in situ observation for temperature dynamics understanding, SST from OISST V2 and OSTIA satellites products are plotted
on each time series. Fig. 6a shows a five-day temperature subset at Uitoe05 station (green curve), moored at 50m depth on the
external slope of the South West lagoon barrier reef in New Caledonia, and the tidal elevation on the same period recomposed
from FES2012 global tide solution (black curve). Temperature drops (by sometimes more than 2°C) are regularly observed at
the M2 tidal wave frequency which suggests the influence of internal tides of high amplitude around New Caledonia
(Bendinger et al., 2023). As expected, the in situ data shows that the satellite data at low and high resolution are neither able
to capture the amplitude observed nor the time scale linked to internal waves illustrating the strong asset of the in situ
observations.  At a similar frequency, the ReefTEMPS time series can also be used to characterise the diurnal temperature
cycle, as depicted in Fig. 6b that displays a two-week temperature series using data from a sensor moored in the Reao atoll
lagoon in 2022. With an offset of more than 1.5°C, the satellite data are not able to capture the level observed in the in-situ
signal. In addition to their primary interest in understanding the physical processes controlling daily and infra-daily temperature
variability, documenting this range of variations may prove useful for benthic species such as coral reefs which can benefit of
some relief during stressing thermal conditions (Wyatt et al., 2020 ; Oliver and Palumbi, 2011). Naturally, daily satellite
products are not able to inform about infra-daily variability but Fig. 6a and 6b also illustrate mean biases introduced when
using such SST products at the coastal scale in coral reef lagoons especially when calculating coral vulnerability indices such
as bleaching indices (Van Wynsberge et al., 2017).
Another key process that can induce significant cooling on the outer slopes of barrier reefs is upwelling. One example is
provided in Fig. 6d where prolonged strong southeasterly trade winds flowing parallel to the coast triggered a wind-driven
coastal upwelling episode in 2021 at station Fausse Passe de Uitoe 05 in New Caledonia, leading to an approx. 4°C decrease
in a few days. This important upwelling feature off the south-west lagoon of New Caledonia can strongly shape biogeochemical
properties of the ocean in the direct vicinity of the lagoon (Alory et al., 2006; Ganachaud et al., 2010). Here again, in situ
observation proves to be essential as satellite SST products fail to reflect the drops of temperature. Finally, Fig. 6d, which
represents a 5-year subset of the temperature time series observed in Fiji (Beqa Island 01 station), highlights the usefulness of
long-term data for understanding seasonal to interannual variability.

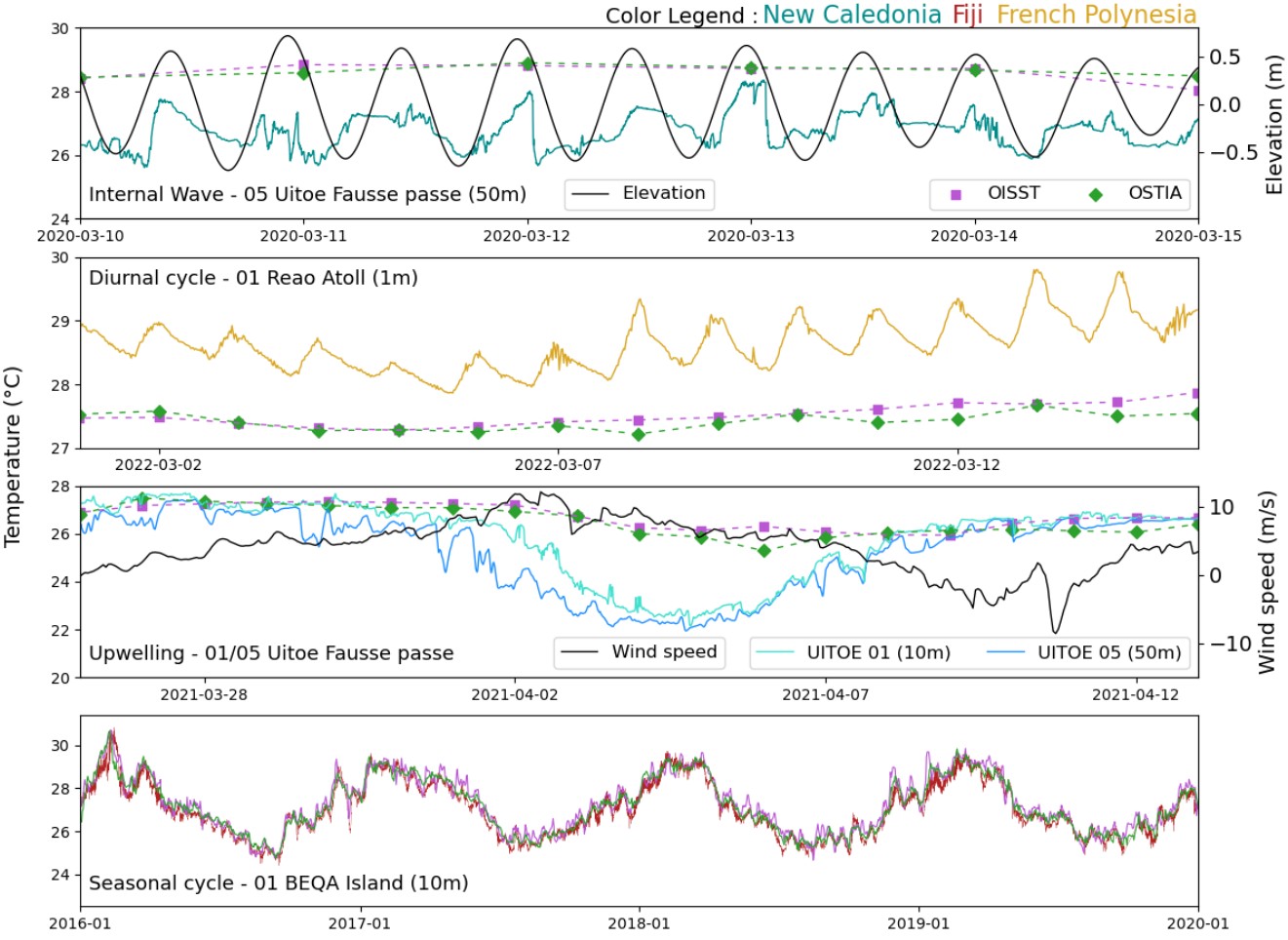


**Figure 6: Illustrations of several typical thermal signatures characterised in situ and using L4 daily satellite products (OISST V2**
**and OSTIA, resp. purple and green points). a. temperature drops due to internal tides at the false passage of Uitoe 05 in New**
**Caledonia (dark cyan curve) and tidal elevation (black curve) recomposed from FES2012 tidal solution at the same station. b.**
**Diurnal cycle at Reao Atoll 01 (yellow curve) in French Polynesia c. upwelling episode at Fausse passe de Uitoe 01 and Uitoe 05**
**stations (resp. 10m and 50m depth). ERA5 wind speed projected along the northeast-southwest main axis of New Caledonia (Figure**
**2) is plotted in a plain curve to illustrate the upwelling event as the wind accelerates on 02/04/2021. d. seasonal and interannual**
**variability of temperature at Beqa island 01 station in Fiji.**
**6.3 Long-term trends**
Some of the historical stations from the ReefTEMPS network date back several decades. These are invaluable observations to
assess the warming trends. Two of these long-term monthly homogenised time series (see 4.2) and associated trends are
presented in Figure 7a and 7b respectively. Both stations, at Anse Vata and Phare Amédée, are located inside the New
Caledonia South-west lagoon but Anse Vata station is very close to the shore whereas Phare Amédée is next to the ocean (see
2.1). Decadal trend computations were performed using Mann-Kendall tests combined with Theil-Sen estimate of linear trend
with the pyMannKendall Python package (Hussain and Mahmud, 2019). The original Mann-Kendall test was used to compute
trends on coldest months and warmer months and the Seasonal Mann-Kendall test on the monthly time series. Considering the
entire observation periods, both stations exhibit increasing trends of 0.125°C / decade and 0.127°C / decade for Anse Vata and
Phare Amédée respectively (p <10e-10 for both tests). Trends calculated using the warmest month of each year do not show
any significant trend for any of the two stations. Conversely, trends on coldest months highlight a significant warming over
the periods with a warming slightly higher next to the ocean (Phare Amédée: 0.185°C / decade) than close to the coast (Anse
Vata : 0.159°C / decade). Finally, it is important to point out that Seager et al. (2022) found, over 5 datasets of global open-
ocean SSTs analysed over 1958-2018, a mean SST trend of ~ 0.1°/decade around New Caledonia (see their Figure 2), which
is weaker than our in-situ trends at Anse Vata and Phare Amédée.

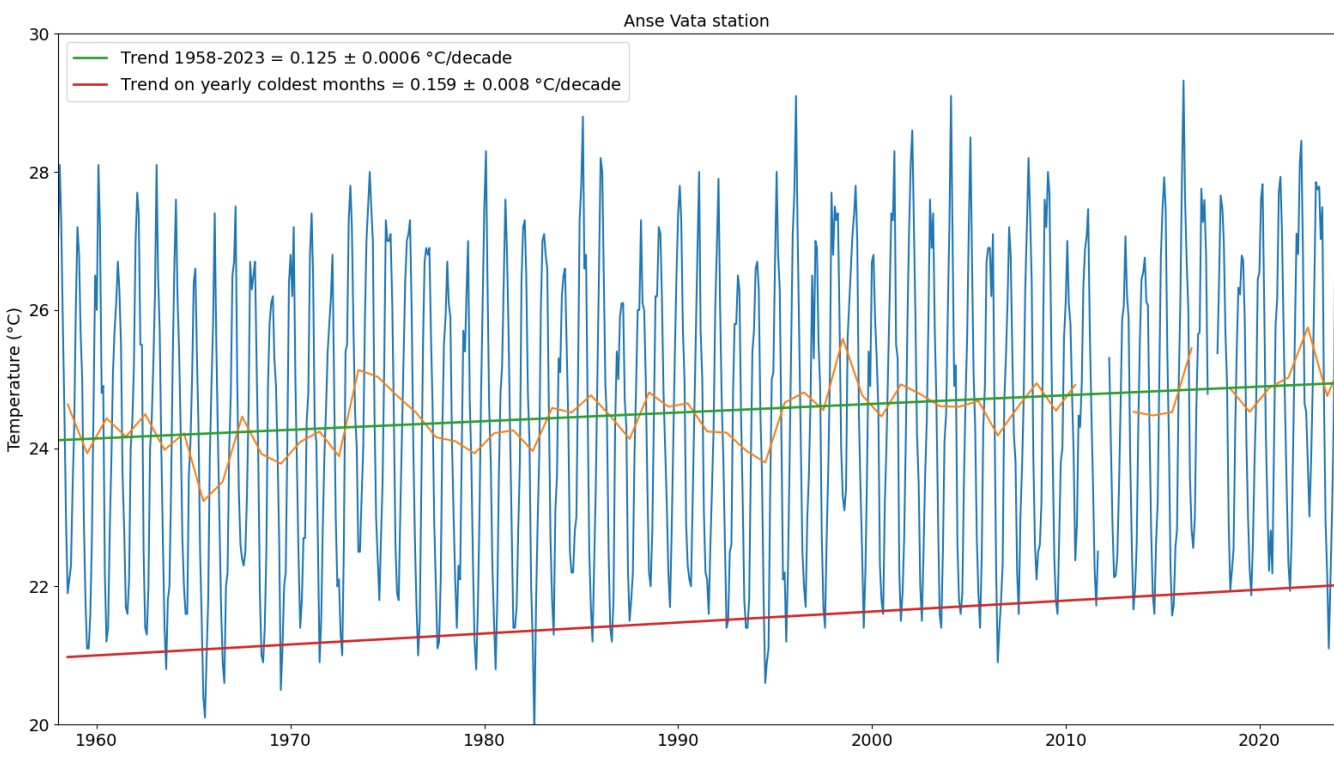


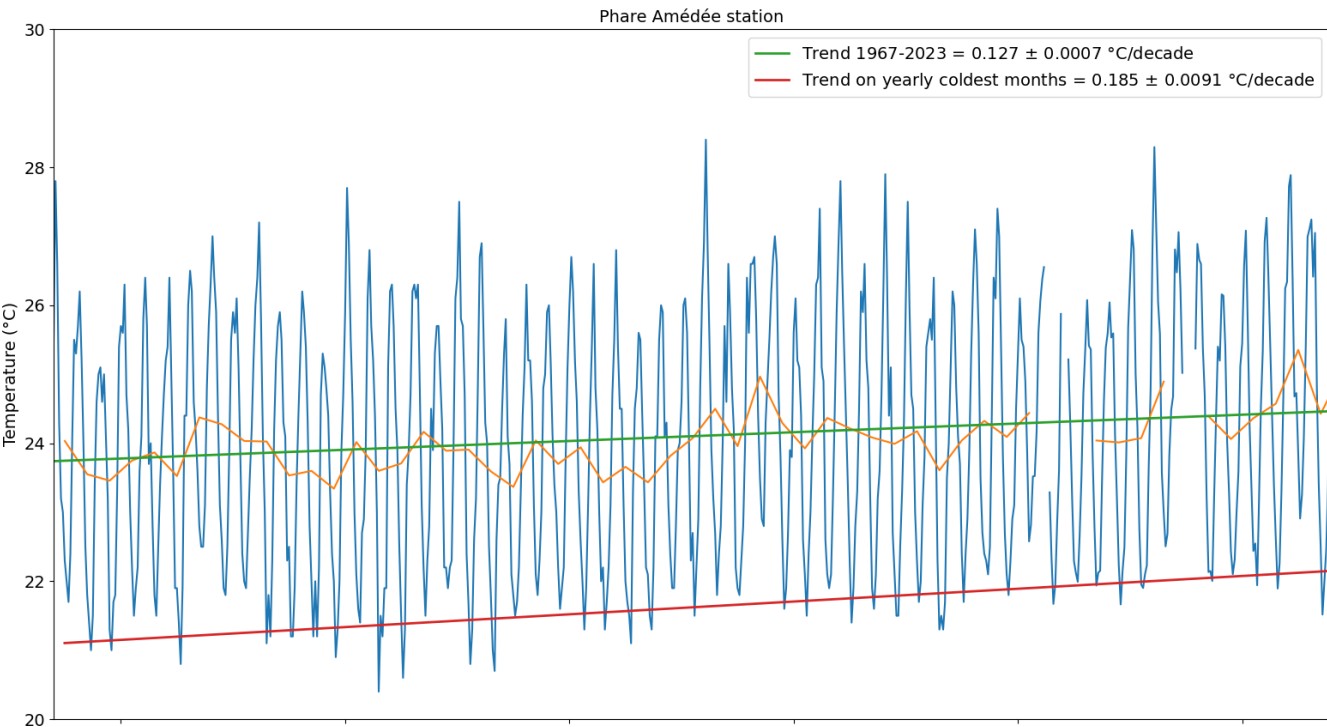

**Figure 7 – Monthly temperature time series and trends at a. Anse Vata station (1958-2023) b. Phare Amédée station (1967-2023). Orange lines are the annual mean time series, green lines trends computed over the whole period, red lines the trend computed on the yearly coldest months over the whole period.**

## 7. Ongoing developments and perspectives

**Technical developments.** As technologies and scientific needs are constantly evolving, the ReefTEMPS consortium develops new functionalities and methods to ensure data robustness, longevity of historical monitoring stations, improved way of disseminating information as well as establishment of new stations.

Concerning the IS and web portal, major evolutions have been underway since 2023 (see section 5) but new developments are still in progress. The next one will concern the data exchange process for which the deployment of OGC SensorThings API will replace the Sensor Observation Service former protocol (see Appendix B). Concerning the workflow manager decision have been made to shift to a workflow manager based on Apache Airflow open-source solution. Using a flow manager has the advantage of being able to adapt easily to the integration of new types of data such as real-time data.

With the increasing threats posed by marine heatwaves on coral reefs, efforts are being put into implementing access to real-time SST observation, which allows informing decision makers on the risks of incoming marine heatwaves. Such systems have already been implemented at Ilot Maître station (see Appendix C). For the first station deployed in New Caledonia at Maitre Island, it consists of an RBR Duet fixed underwater to a pile of one of the bungalows of the Hilton hotel and connected by an electronic cable to a Raspberry-type nanocomputer equipped with a LoRa transmission antenna. The measurements are

recorded on a memory card on the Raspberry and sent in packets every 15 minutes by LoraWan transmission. A Lora receiver
within radio range of the station recovers the data and transmits it over the internet. It is then recovered by the ReefTEMPS
information system and processed into the database. Two strategies are envisioned for the future deployment of such real-time
array:
1- A low-cost strategy whenever possible using Internet of Things (IoT) communication technology (Mattern and
Floerkemeier, 2010): a new station with such technology will be implemented at Phare Amédée during 2024.
2- A regular strategy with 4G or Iridium transmission for stations where IOT cannot be implemented.
Figure 8 presents the beta version of live data at Ilot Maitre available on the ReefTEMPS data portal web page. New
developments are underway to display visualisation of real-time indicators such as Degree Heating Week index (DHW) used
in many instances to indicate a risk for coral bleaching (ref) or Marine Heatwave real-time information. These potential
applications of real-time SST data can be crucial for public bodies and researchers to access information crucial for lagoon
ecosystems vulnerability in terms of preparedness and management.

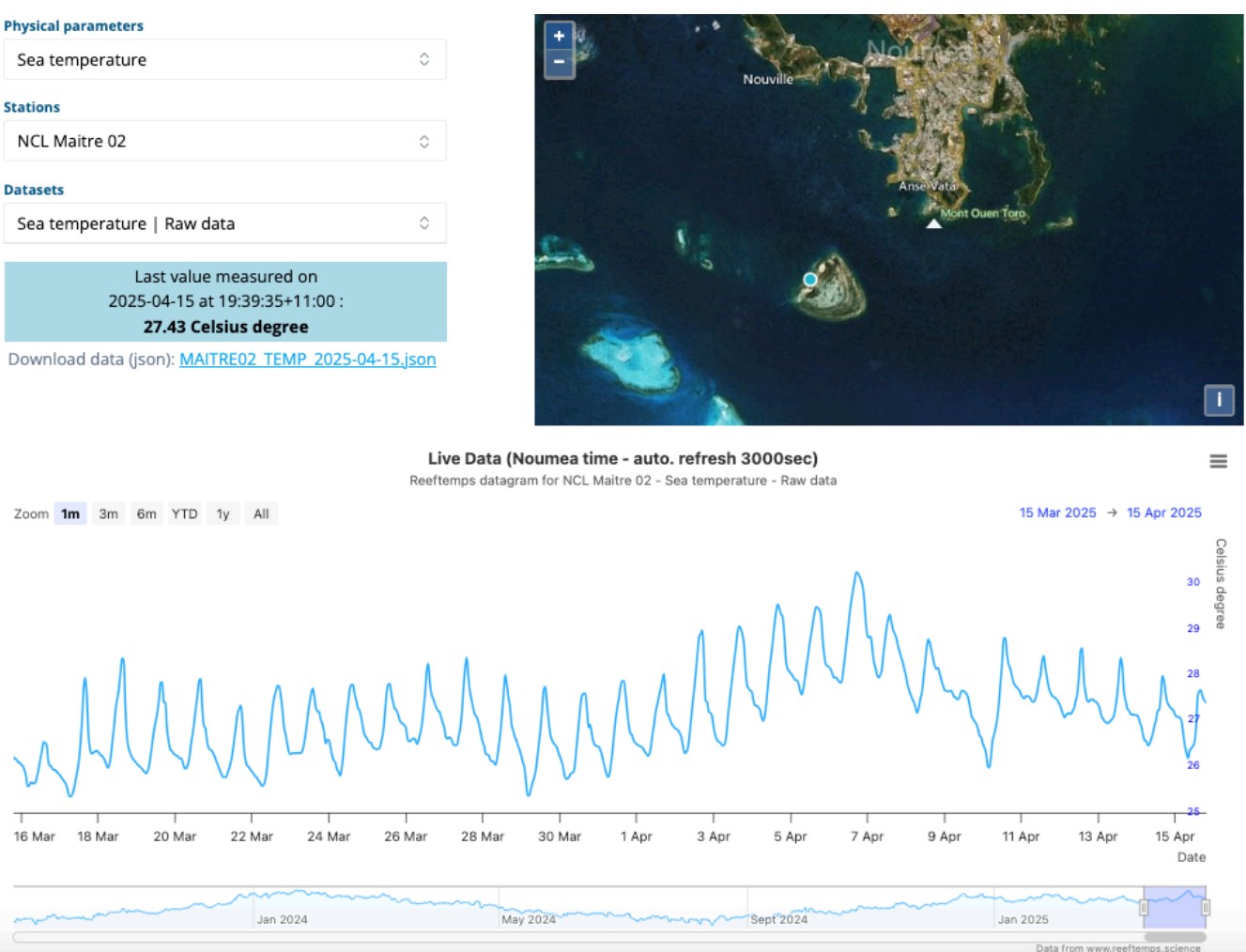


**Figure 8 – Beta version of the live data webpage available on the ReefTEMPS. The temperature time series is plotted at the blue**
**point (Ilot Maitre live station) on the map (https://www.reeftemps.science/en/live/).**

**Quality perspectives**. To ensure more rapid and robust controls of accuracy of observations given the remote locations of the
Pacific Countries to the instrument manufacturers, two new strategies have been introduced. First, a local SBE56
intercomparison protocol has been developed recently (Detandt and Varillon, 2024). The principle is that all SBE56 sensors
are inter-compared in a temperature-controlled tank (from 20 to 32°C) with a reference sensor that has recently returned from
calibration at the manufacturer (Seabird). The measured differences to the reference sensor must not exceed +/- 0.005°C from
the calibrated sensor or that sensor is sent to the manufacturer for calibration. For stations in front of Nouméa, a second strategy
that will enable a robust data quality control as well as a characterisation of the water column is a monthly visit of several
stations to perform profiles over the water column with a calibrated SBE19PlusV2 CTD. Concerning long-term time series, a
work has recently begun to convert raw data into daily (so far monthly) homogenised data to ensure a perfect reliability for
computing trends and climate induced warming. Finally, the Python tool developed for the qualification will need to be
continued, to provide a fully integrated tool that works directly on the DB-Oceano database and provides more statistical
functionalities.
**Future strategies for site selection.** In all monitored PICTs, future strategy will first focus on maintaining historical long-
term stations to provide a spatial view of the warming trends. In New Caledonia, the choice of new station locations will be
guided to deeper investigate the signal deformation between open ocean temperature and lagoon temperature. In Fiji, efforts
will be made to maintain unbroken time-series for the longest-established sites, such as Suva Reef and Rotuma, while
expanding to new sites in the Lau Group and Vanua Levu. A major challenge has been the closing of some sites such as Batiki,
Coral Coast and Yasawa, due to the loss of local partners. Another issue is the damage or loss of monitoring platforms (Beqa,
Batiki) due to seasonal tropical cyclones. Furthermore, the ReefTEMPS working group also plans to deploy more vertical
arrays on the external barrier reefs slopes for a more thorough understanding of the processes leading to cooling (e.g internal
waves, upwelling). Future observation sites in French Polynesia will mainly be dedicated to important pearl farming atolls. In
Fiji, such a pilot vertical array to 200m depth had already been deployed on a mooring off the Coral Coast of Viti Levu Island,
as part of the VERTEMP Project under the IRD JEAI COPRA between May to November 2018, and January 2019 to January
2020, sampling at 30 second intervals with an array of ten SBE56 sensors (N'Yeurt et al., in prep.). Finally, the coastal
monitoring sites on Wallis and Futuna will also be re-instrumented in the near future.
**Diversifying observations and ocean in ReefTEMPS.**
At present, ReefTEMPS is mostly based on an array of temperature sensors but the increasing challenge of long-term coastal
observations is to couple these measurements with other key measurements such as salinity, pressure sensors for coastal
vulnerability issues and biogeochemistry (e.g pH, fluorescence, turbidity, nutrients phytoplankton pigments, etc…) to monitor
water quality and ecosystems. In our studies of the coral reef environment and bleaching surveillance, we perform regular in
situ campaigns crossing ReefTEMPS stations with suites of physical and biological punctual measurements. The long-term
plan for our coastal observing system is to systematically add to the automated temperature array, other automated sensors to
provide a more complete monitoring of the environment facing climate change. Along these lines, a long-term, reliable funding
system has to be secured, a key challenge that will require strong involvement of the government agencies for which these
measurements are performed and that are lacking at present. Nevertheless, even if this paper targeted temperature observations,
ReefTEMPS is also labelled by SNO (see 2.1) for other observables such conductivity and pressure. Therefore, in addition to
temperature, conductivity and pressure time series are also available on many stations through the ReefTEMPS open database.
Some other key in situ time series have also started in New Caledonia: pH continuous observations using in situ sensors for
example at Fausse Passe de Uitoé or waves using spotter buoys. In Fiji, an experimental autonomous spectrophotometry-based
pH sensor had been deployed on several occasions at the VELEVU02 site near Suva in collaboration with the National
Oceanography Centre (NOC) of the United Kingdom, and preliminary datasets uploaded on the IOC-UNESCO SDG 14.3.1

data portal (https://oa.iode.org/, last access: 5 September 2024). It is hoped to continue such observations of in situ pH at this and other ReefTEMPS sites in Fiji. Finally, ReefTEMPS environment, quality observation and practices are now extended to the Indian Ocean in La Réunion Island as a first step with the will to continue that collaborative effort with other Indian Ocean Countries.

**8. Data availability**

All station time series are available individually, either in ASCII or NetCDF formats, on the ReefTEMPS web portal: https://www.reeftemps.science/. The whole ReefTEMPS dataset is freely available in NetCDF format on the dedicated SEANOE repositories (Varillon et al., 2025: https://doi.org/10.17882/55128 ; Liao et al., 2025: https://doi.org/10.17882/82291) and updated every semester. Filenames, variable names, dimensions, and attributes, as well as quality flagging, follow international standards, in particular the OceanSITES and NERC Vocabulary Server conventions (see Section 5 and Appendix A).

**9. Conclusion**

The ReefTEMPS network presented in this paper represents a unique source of knowledge for understanding coastal temperature, salinity and pressure dynamics in the South Pacific Ocean and for monitoring coral reef ecosystem thermal variability. The most striking feature that makes this network unique and extremely valuable is undoubtedly its geographical coverage (16 PICTS covered, 115 stations monitored) of temperature sensors and the duration of observations for some of its oldest monitoring stations (since 1958 for Anse Vata in New Caledonia). The network ensures open access and quality controlled in situ data that can be visualised and downloaded through the internet in ASCII and NetCDF formats according to the FAIR principles. Usefulness of these data is considerable as they can be used to investigate coastal and lagoon processes on different time scales such as waves dynamics, upwelling, extreme marine heatwave events, tropical cyclone impacts, long term interannual to decadal variabilities and climate warming trends. The basin-scale distribution of the ReefTEMPS network is also crucial for accurately capturing the spatially heterogeneous impacts of large-scale climate phenomena such as the El Niño–Southern Oscillation or the Pacific Decadal Oscillation. Finally, this in-situ network is a key asset for validating the development of remotely-sensed observations, which, at present, cannot represent the fine-scale, high temporal resolution depicted by the ReefTEMPS network and these data can be used for ocean model tuning and evaluations. In addition to highlighting the scientific value of the ReefTEMPS dataset, this paper aimed at bringing the ReefTEMPS network to the attention of as many researchers as possible and inviting interested partners from the Pacific Island Countries and Territories to join the initiative.

## Author contributions

RLG, ALS, CM, SC, SF and RH prepared the paper and designed the figures with contributions from all co-authors. All the co-authors have been strongly involved in the ReefTEMPS network at some points in its life (in situ operations, web portal, organisation, processing and checking of data) or helped to raise funds to support it.

## Competing interests

The contact author has declared that none of the authors has any competing interests.

## Acknowledgement

We would like to provide here our warmest thanks to all the researchers and technical staff (operators, scuba divers, boat drivers…) who have contributed to the success of this network through the years, with special attention to the historical IRD team in IRD Nouméa with special thanks to Pierre Waigna, Christian Hénin who greatly contributed to the development of the initial network. Our sincere thanks also go to the entire UAR IMAGO over the years, and especially Céline Bachelier, Damien Vignon, Guillaume Detandt, which have contributed so much to the continuity of this network. Discussions with G. Reverdin on bucket sampling are also acknowledged.

## Financial Support

Financial support has evolved over six decades but came mainly from the institutes themselves (ORSTOM/ the the French National Research Institute for Sustainable Development (IRD)/ The University of South Pacific (USP) / The Pacific Community (SPC), the Marine Resources Department of the French Polynesia (DRM)), with the support of external resources (the ZONECO project of the New Caledonian Government, Ministère de l'Outre-Mer Français, the GOPS (Grand Observatoire du Pacifique Sud)), and finally a long lasting and national support: the French Infrastructure for Coastal Oceans and Seashores ILICO, with the French Ministry of Higher Education (MESR), Research and the CNRS-INSU.

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

**Appendix A**

**Table A1. Countries and Territories codes**

| Country | Code | Country | Code |
|---------|------|---------|------|
| Cook Islands | COK | Niue | NIU |
| Micronesia | FSM | Palau | PLW |
| Fiji | FJI | Papua New Guinea | PNG |
| French Polynesia | PYF | Samoa | WSM |
| Kiribati | KIR | Tokelau | TKL |
| Marshall Islands | MHL | Tuvalu | TUV |
| Nauru | NRU | Vanuatu | VUT |
| New Caledonia | NCL | Wallis and Futuna | WLF |

**Table A2. Instrument types (Fiat et al. 2024) derived from NERC L05 ( https://vocab.nerc.ac.uk/collection/L05/current/)**

| Code | Label | Description |
|------|-------|-------------|
| TR | THERMISTOR | THERMISTOR OR THERMISTOR CHAIN |
| CT | CTD | CONDUCTIVITY TEMPERATURE DEPTH (CTD) PROBE |
| TS | TSG | THERMOSALINOGRAPH |
| MG | TIDEGAUGE | TIDEGAUGE |
| SM | BUCKET | METEOROLOGICAL BUCKET |
| MP | MULTIPARAMETER PROBE | MULTIPARAMETER PROBE |
| PH | PHMETER | PH METER |

**Table A3. Processing states (Fiat et al. 2024) derived from NERC Vocabulary Server (NVS) R06 (https://vocab.nerc.ac.uk/collection/R06/current/)**


| Processing level | ReefTEMPS Description |
|---|---|
| 0A | RAW DATA |
| 1A | GEOPHYSICAL UNCHECKED DATA |
| 2B | QUALITY-CONTROLLED DATA |
| 2C | HIGHEST QUALITY REFERENCED DATA |
| 3A | UNCHECKED TIME RESAMPLED DATA |
| 3B | QUALITY CONTROLLED TIME RESAMPLED DATA |
| 3C | PEER-REVIEWED TIME RESAMPLED REFERENCED DATA |


**Table A4. ReefTEMPS stations informations: Names, positions, sensor type, depths, start and end dates, total duration (days)**

| Station Name | Station Code | Longitude | Latitude | Sensor Type | Depth | Active | Start | End | Duration (days) |
|---|---|---|---|---|---|---|---|---|---|
| NCL Fausse passe de Uitoe 01 | UITOE01 | 166.1832 | -22.2859 | TS | 10.0 | True | 1992-05-22 | 2024-07-30 | 11757 |
| NCL Récif du Prony 01 | RECPRO01 | 166.3325 | -22.2673 | TR | 10.5 | True | 1996-01-12 | 2024-01-10 | 10426 |
| NCL Anse Vata 01 | ANSEVA01 | 166.4433 | -22.3038 | TR | 2 | True | 1997-04-15 | 2024-08-26 | 9995 |
| NCL Phare Amedee 01 | PHARAM01 | 166.466 | -22.4757 | TR | 4.5 | True | 1997-06-01 | 2024-01-15 | 9933 |
| NCL Chesterfield 01 | CHESTE01 | 158.3076 | -19.8747 | TR | 17 | True | 1997-09-24 | 2024-11-01 | 9899 |
| NCL Surprises 01 | SURPRI01 | 163.0781 | -18.4853 | TR | 14 | True | 1997-09-28 | 2024-03-17 | 9667 |
| NCL Goro 01 | GORO01 | 167.1072 | -22.2725 | TR | 11 | True | 1997-04-03 | 2023-06-18 | 9572 |
| NCL Poe Beach 01 | BOURAI01 | 165.3388 | -21.6123 | TR | 4 | True | 1999-08-12 | 2024-02-25 | 8963 |
| NCL Poindimié 01 | POINDI01 | 165.4850 | -20.8918 | TR | 12.5 | True | 1997-10-22 | 2023-10-29 | 5851 |
| NCL Koumac 01 | KOUMAC01 | 164.1901 | -20.6636 | TR | 14 | True | 2011-01-28 | 2024-11-07 | 5032 |
| NCL Canard 01 | CANARD01 | 166.4339 | -22.3122 | TR | 5 | True | 2011-01-19 | 2024-08-26 | 4968 |
| NCL Ouano 02 | UARAI01 | 165.7238 | -21.8616 | TR | 12 | True | 2011-11-30 | 2024-11-08 | 4727 |
| NCL Maitre 01 | MAITRE01 | 166.4030 | -22.3417 | TS | 3.5 | True | 2012-04-26 | 2024-07-29 | 4476 |
| NCL Ouano 01 | CHAMBE01 | 165.7861 | -21.8170 | TR | 9 | True | 2011-11-30 | 2024-01-09 | 4423 |
| FJI Viti Levu Island 01 | VELEVU01 | 178.5140 | -17.5220 | TR | 12 | True | 2012-11-30 | 2023-12-13 | 4030 |

| | | | | | | | | | |
|---|---|---|---|---|---|---|---|---|---|
| FJI Viti Levu Island 02 | VELEVU02 | 178.3999 | -18.1597 | TR | 12 | True | 2012-12-21 | 2023-11-17 | 3983 |
| NCL Poindimié 01 | POINDI01 | 165.4850 | -20.8918 | MG | 12.5 | True | 2013-10-31 | 2024-08-18 | 3944 |
| NCL Poindimié 02 | POINDI02 | 165.3220 | -20.9288 | MG | 1.7 | True | 2013-09-17 | 2024-03-03 | 3820 |
| FJI Rotuma Island 01 | ROTUMA01 | 177.0432 | -12.5199 | TR | 12 | True | 2014-09-18 | 2024-11-13 | 3709 |
| NCL Ile des pins 01 | IDPINS01 | 167.4352 | -22.5287 | TR | 14 | True | 2015-09-30 | 2024-08-21 | 3248 |
| NCL Ile des pins 02 | IDPINS02 | 167.3509 | -22.6490 | TR | 13 | True | 2015-10-05 | 2024-08-22 | 3243 |
| NCL Baie des citrons 01 | LEMONB01 | 166.4353 | -22.2958 | TS | 3.0 | True | 2016-02-26 | 2024-07-29 | 3076 |
| NCL Fausse Passe de Uitoe 04 | UITOE04 | 166.1930 | -22.2859 | MG | 20 | True | 2016-06-23 | 2024-07-30 | 2958 |
| NCL Ilot Laregnere 01 | LAREGN01 | 166.3198 | -22.3311 | MG | 8 | True | 2016-06-23 | 2024-07-17 | 2946 |
| NCL Ilot Mbe-Kouen 01 | MBEKOU01 | 166.2213 | -22.2677 | MG | 6.5 | True | 2016-06-23 | 2024-07-17 | 2946 |
| FJI Vatu-i-Ra Passage 01 | VATUIR01 | 178.5930 | -17.3315 | TR | 9.5 | True | 2016-12-04 | 2024-02-20 | 2634 |
| WLF Alofi island 01 | ALOFI01 | -178.074 | -14.3371 | TR | 11 | True | 2012-10-18 | 2019-10-30 | 2567 |
| FJI BEQA Island 01 | BEQA01 | 178.1675 | -18.4137 | TR | 10 | True | 2014-05-28 | 2020-11-06 | 2354 |
| NCL Récif de Basse Kaui 01 | BAKAUI01 | 166.3159 | -22.2466 | CT | 8 | True | 2013-08-13 | 2019-05-19 | 2104 |
| PYF Mangareva Atoll 01 | MANGAR01 | -135.0048 | -23.0902 | MP | 3.5 | True | 2018-05-24 | 2023-12-13 | 2029 |
| FJI BEQA Island 02 | BEQA02 | 178.1956 | -18.3769 | TR | 12 | True | 2014-05-28 | 2019-09-26 | 1947 |
| NCL Ilot Redika 01 | REDIKA01 | 166.6104 | -22.5191 | MG | 11.5 | True | 2018-10-05 | 2024-01-30 | 1943 |
| NCL Fausse passe de Uitoe 05 | UITOE05 | 166.1832 | -22.2859 | TR | 50 | True | 2019-07-25 | 2024-08-05 | 1838 |
| PYF Takapoto Atoll 01 | TAKAPO01 | -145.2456 | -14.7037 | TR | 3 | True | 2020-08-08 | 2023-11-15 | 1194 |
| PYF Arutua Atoll 01 | ARUTUA01 | -146.6167 | -15.2646 | MP | 3.5 | True | 2018-06-15 | 2021-07-23 | 1134 |
| NCL Hienghene 01 | HIENGE01 | 164.9839 | -20.6449 | TS | 3 | True | 2022-01-01 | 2024-12-18 | 1082 |
| PYF Reao Atoll 01 | REAO01 | -136.4248 | -18.4830 | TR | 1 | True | 2021-06-21 | 2024-04-01 | 1015 |
| NCL Recif Snark 01 | SNARK01 | 166.4263 | -22.4437 | TR | 3 | True | 2022-01-31 | 2024-08-28 | 940 |
| NCL Passe Boulari 02 | BOULAR02 | 166.4304 | -22.4842 | TR | 3 | True | 2022-02-01 | 2024-08-28 | 939 |
| NCL Passe Boulari 03 | BOULAR03 | 166.4320 | -22.4907 | TR | 6.5 | True | 2022-01-31 | 2024-08-28 | 939 |
| NCL Koumac 01 | KOUMAC01 | 164.1901 | -20.6636 | TS | 14 | True | 2008-07-22 | 2010-08-29 | 768 |
| PYF Tahaa Atoll 01 | TAHAA01 | -151.5562 | -16.5954 | TR | 3.5 | True | 2021-06-18 | 2023-07-21 | 763 |
| PYF Takaroa Atoll 04 | TAKARO04 | -144.9595 | -14.4597 | MP | 4 | True | 2019-01-30 | 2021-02-24 | 756 |

| | | | | | | | | | |
|---|---|---|---|---|---|---|---|---|---|
| NCL Chesterfield 02 | CHESTE02 | 158.6062 | -19.2143 | TR | 5,5 | True | 2022-10-13 | 2024-11-05 | 754 |
| PYF Arutua Atoll 01 | ARUTUA01 | -146.6167 | -15.2646 | TR | 3.5 | True | 2021-12-01 | 2023-11-07 | 706 |
| NCL Touho 02 | TOUHO02 | 165.2439 | -20.7700 | TR | 1 | True | 2022-12-01 | 2024-09-05 | 644 |
| FJI Vanuabalavu Island 01 | BALAVU01 | 179.0630 | -17.1450 | TR | 10 | True | 2022-08-07 | 2024-03-18 | 589 |
| NCL Atoll de Huon 01 | HUON01 | 162.8285 | -18.0708 | TR | 4 | True | 2022-10-14 | 2024-03-15 | 517 |
| NCL Fausse passe de Uitoe 01 | UITOE01 | 166.1832 | -22.2859 | PH | 11 | True | 2022-09-02 | 2024-01-10 | 495 |
| NCL Lifou Island 01 | LIFOU01 | 167.12108 | -20.78875 | TR | 5.5 | True | 2023-07-18 | 2024-11-22 | 492 |
| NCL Récif de Basse Kaui 01 | BAKAUI01 | 166.3159 | -22.2466 | TR | 8 | True | 2023-04-24 | 2024-07-29 | 461 |
| FJI ONO-I-LAU Island 01 | ONOILO01 | -178.7512 | -20.6220 | TR | 12 | True | 2021-12-06 | 2023-02-22 | 443 |
| PYF Tahaa Atoll 01 | TAHAA01 | -151.5562 | -16.5954 | MP | 3.5 | True | 2018-10-17 | 2019-11-03 | 382 |
| NCL Chesterfield 03 | CHESTE03 | 158.4580 | -19.9505 | TR | 4,5 | True | 2023-10-28 | 2024-10-30 | 368 |
| FJI Vulaga Island 01 | VULAGA01 | -178.5569 | -19.1396 | TR | 8.2 | True | 2022-08-20 | 2023-08-17 | 362 |
| PYF Takapoto Atoll 01 | TAKAPO01 | -145.2456 | -14.7037 | MP | 3 | True | 2020-02-06 | 2020-10-31 | 267 |
| PYF AHE Atoll 01 | AHE01 | -146.3791 | -14.5263 | MP | 2 | True | 2022-03-23 | 2022-09-18 | 178 |
| FJI Vulaga Island 02 | VULAGA02 | -178.5400 | -19.1213 | TR | 12.4 | True | 2022-08-20 | 2022-12-20 | 122 |
| NCL Hienghene 01 | HIENGE02 | 165.0035 | -20.5626 | TR | 5 | True | 2022-11-29 | 2023-03-27 | 118 |
| NCL Hienghene 01 | HIENGE02 | 165.0035 | -20.5626 | TR | 27 | True | 2022-11-29 | 2023-03-27 | 118 |
| PYF Vairao 01 | VAIRAO01 | -149.2933 | -17.8064 | TR | 3 | True | 2023-05-19 | 2023-08-04 | 77 |
| NCL Anse Vata 01 | ANSEVA01 | 166.4433 | -22.3038 | SM | 0.5 | False | 1958-07-18 | 2005-06-28 | 17747 |
| NCL Phare Amedee 01 | PHARAM01 | 166.4660 | -22.4757 | SM | 0.5 | False | 1967-02-28 | 2000-09-29 | 12266 |
| NCL Passe Boulari 01 | BOULAR01 | 166.4317 | -22.4917 | TR | 14 | False | 1996-01-11 | 2024-08-28 | 10457 |
| PYF Tahiti 01 | TAHITI01 | -149.5679 | -17.5213 | SM | 0.5 | False | 1979-01-04 | 1989-05-30 | 7667 |
| NCL Fausse passe de Uitoe 03 | UITOE03 | 166.1832 | -22.2859 | TR | 60.0 | False | 2001-07-23 | 2021-10-29 | 7403 |
| PYF Marquises 01 | NUKUHI01 | -140.0944 | -8.9342 | TR | 10 | False | 1997-09-19 | 2010-11-21 | 4811 |
| NCL Fausse passe de Uitoe 01 | UITOE01 | 166.1832 | -22.2859 | TR | 11 | False | 1999-09-20 | 2010-06-20 | 3925 |
| VUT Wusi 01 | WUSI01 | 166.5681 | -15.3702 | MG | 11 | False | 1999-11-19 | 2010-05-29 | 3844 |
| VUT Sabine 01 | SABINE01 | 166.1362 | -15.9467 | MG | 11 | False | 1999-11-18 | 2010-05-26 | 3842 |
| NCL Fausse passe de Uitoe 02 | UITOE02 | 166.1832 | -22.2859 | TR | 30.0 | False | 2001-07-23 | 2010-06-20 | 3254 |

| WLF Wallis 02 | WALLIS02 | -176.2767 | -13.3091 | TR | 10 | False | 2006-10-17 | 2015-08-27 | 3235 |
|---|---|---|---|---|---|---|---|---|---|
| FSM Pohnpei 02 | POHNPE02 | 158.1119 | 6.8001 | TR | 13 | False | 2010-10-01 | 2018-10-15 | 2936 |
| FSM Pohnpei 01 | POHNPE01 | 158.2969 | 7.0093 | TR | 13 | False | 2010-10-01 | 2018-09-28 | 2919 |
| NCL Belep 01 | BELEP01 | 163.6450 | -19.7156 | SM | 0.5 | False | 1978-06-09 | 1986-05-30 | 2912 |
| FJI Batiki Island 01 | BATIKI01 | 179.1799 | -17.7775 | TR | 10 | False | 2012-11-28 | 2019-01-25 | 2249 |
| MHL Majuro 03 | MAJURO03 | 171.0542 | 7.1924 | TR | 9 | False | 2012-08-25 | 2018-07-31 | 2166 |
| PYF Vairao 01 | VAIRAO01 | -149.2933 | -17.8064 | MP | 3 | False | 2018-03-03 | 2023-05-19 | 1903 |
| NCL Ouvéa 02 | OUVEA02 | 166.4882 | -20.6533 | MG | 8 | False | 2013-09-23 | 2018-08-28 | 1800 |
| WLF Wallis 01 | WALLIS01 | -176.2516 | -13.2222 | TS | 11 | False | 1998-08-21 | 2003-03-16 | 1667 |
| FJI Viti Levu Island 03 | VELEVU03 | 177.6732 | -18.2100 | TR | 11.9 | False | 2013-04-05 | 2017-09-23 | 1632 |
| FJI Batiki Island 02 | BATIKI02 | 179.1390 | -17.7855 | TR | 10 | False | 2012-11-29 | 2017-03-16 | 1568 |
| VUT Sabine 01 | SABINE01 | 166.1362 | -15.9467 | TR | 11 | False | 1999-11-18 | 2004-02-29 | 1564 |
| VUT Santo Island 01 | SANTO01 | 167.2798 | -15.5480 | TR | 8 | False | 2012-06-25 | 2016-05-15 | 1420 |
| NCL Passe de Dumbea 01 | DUMBEA01 | 166.1887 | -22.2957 | TR | 9 | False | 1996-01-10 | 1999-09-19 | 1348 |
| NCL Passe de Dumbea 02 | DUMBEA02 | 166.2688 | -22.3705 | TR | 11 | False | 1996-01-10 | 1999-09-19 | 1348 |
| NCL Nouville 01 | NOUVIL01 | 166.4182 | -22.2782 | TR | 11 | False | 1996-01-12 | 1999-09-20 | 1346 |
| FJI Tawewa Island 01 | TAWEWA01 | 177.3675 | -16.9221 | TR | 10 | False | 2012-12-08 | 2016-05-15 | 1254 |
| PYF Hapou 01 | HAPOU01 | -140.0468 | -9.3571 | SM | 0.5 | False | 1986-01-31 | 1989-06-14 | 1230 |
| PYF Takaroa Atoll 01 | TAKARO01 | -145.0161 | -14.5026 | TR | 4 | False | 2012-11-29 | 2016-03-20 | 1207 |
| FJI Tawewa Island 02 | TAWEWA02 | 177.3379 | -16.8806 | TR | 16.2 | False | 2012-12-08 | 2016-02-09 | 1158 |
| VUT Efate Island 01 | EFATE01 | 168.2632 | -17.7696 | TR | 8 | False | 2012-06-20 | 2015-08-20 | 1156 |
| PYF Takaroa Atoll 03 | TAKARO03 | -145.0524 | -14.5076 | TR | 2 | False | 2012-11-29 | 2016-01-27 | 1154 |
| NCL Le Cap Goulvain 02 | LECAP02 | 165.2461 | -21.5668 | TR | 20.5 | False | 2012-08-16 | 2015-08-06 | 1085 |
| NCL Le Cap Goulvain 03 | LECAP03 | 165.2397 | -21.5359 | TR | 18 | False | 2012-08-15 | 2015-08-05 | 1085 |
| NCL Le Cap Goulvain 04 | LECAP04 | 165.2413 | -21.5250 | TR | 1.8 | False | 2012-08-15 | 2015-08-05 | 1085 |
| NIU Niue Island 01 | NIUE01 | -169.9192 | -19.0449 | TR | 15 | False | 2016-09-29 | 2019-08-22 | 1057 |
| PNG Manus 02 | MANUS02 | 147.0964 | -1.9318 | TR | 12 | False | 2011-07-31 | 2014-05-11 | 1014 |
| PNG Manus 01 | MANUS01 | 147.0965 | -1.9450 | TR | 10 | False | 2011-07-31 | 2014-05-07 | 1010 |

| | | | | | | | | | |
|---|---|---|---|---|---|---|---|---|---|
| PYF Rapa 01 | RAPA01 | -144.3323 | -27.618 | SM | 0.5 | False | 1986-05-09 | 1989-01-29 | 996 |
| VUT Wusi 02 | WUSI02 | 166.6602 | -15.355 | MG | 11 | False | 2007-10-14 | 2010-05-30 | 958 |
| PYF Tatakoto Atoll 01 | TATAKO01 | -138.4353 | -17.3488 | TR | 1 | False | 2012-11-07 | 2015-06-14 | 949 |
| PYF Tatakoto Atoll 02 | TATAKO02 | -138.3513 | -17.3334 | TR | 2.2 | False | 2012-11-09 | 2015-06-15 | 947 |
| WSM Upolu 01 | UPOLU01 | -172.1281 | -13.8455 | TR | 10 | False | 2012-08-31 | 2015-03-30 | 941 |
| NCL Saint Vincent 01 | STVINC01 | 166.0814 | -21.9271 | TS | 0.5 | False | 2021-12-21 | 2024-07-11 | 933 |
| WLF Wallis 01 | WALLIS01 | -176.2516 | -13.2222 | TR | 10 | False | 2003-03-16 | 2005-09-14 | 912 |
| COK Manihiki Atoll 01 | MHXCOK01 | -160.9969 | -10.4238 | TR | 5.0 | False | 2012-10-27 | 2015-01-25 | 820 |
| COK Manihiki Atoll 02 | MHXCOK02 | -160.9969 | -10.4238 | TR | 20.0 | False | 2012-10-27 | 2015-01-25 | 820 |
| NCL Le Cap Goulvain 01 | LECAP01 | 165.2378 | -21.5529 | TR | 10.0 | False | 1997-03-09 | 1999-05-27 | 809 |
| TKL Nukunonu 01 | NUKUNN01 | -171.8522 | -9.2007 | TR | 8 | False | 2012-05-04 | 2014-05-22 | 747 |
| PYF Tatakoto Atoll 03 | TATAKO03 | -138.4099 | -17.3508 | MG | 1.9 | False | 2012-11-13 | 2014-10-24 | 709 |
| NCL Ouvéa 01 | OUVEA01 | 166.5610 | -20.5489 | MG | 2 | False | 2013-09-23 | 2015-08-19 | 691 |
| FSM YAP 01 | YAP01 | 138.1411 | 9.5030 | TR | 9 | False | 2012-12-14 | 2014-09-21 | 646 |
| TUV Funafuti 01 | FUNAFU01 | 179.0601 | -8.4850 | TR | 11 | False | 2011-08-01 | 2013-04-19 | 627 |
| TUV Funafuti 02 | FUNAFU02 | 179.1328 | -8.5638 | TR | 4 | False | 2011-08-15 | 2013-04-24 | 618 |
| PYF Tubuai Island 03 | TUBUAI03 | -149.4141 | -23.4044 | MG | 1.5 | False | 2013-04-27 | 2014-12-02 | 584 |
| NCL Sainte Marie 01 | SMARIE01 | 166.4813 | -22.3037 | TR | 4.4 | False | 2012-02-03 | 2013-04-22 | 444 |
| TKL Nukunonu 02 | NUKUNN02 | -171.8475 | -9.2007 | TR | 12 | False | 2012-05-05 | 2013-07-09 | 429 |
| COK Manihiki Atoll 03 | MHXCOK03 | -160.9969 | -10.4238 | MG | 15.0 | False | 2012-10-29 | 2013-12-21 | 417 |
| VUT Vanua Lava Island 01 | VANULA01 | 167.5648 | -13.8673 | TR | 5 | False | 2012-06-27 | 2013-07-11 | 378 |
| NRU Nauru 01 | NAURU01 | 166.9537 | -0.5300 | TR | 9.5 | False | 2012-06-18 | 2013-06-23 | 370 |
| NCL Mato 01 | MATO01 | 166.7896 | -22.5597 | TR | 10 | False | 2004-12-09 | 2005-12-08 | 363 |
| NCL Récif Ngedembi 01 | NGEDEM01 | 167.0373 | -22.9688 | TR | 14 | False | 2004-12-10 | 2005-12-07 | 361 |
| NCL Ilot NDA 01 | ILONDA01 | 166.8764 | -22.8497 | MG | 11 | False | 2019-09-21 | 2020-08-12 | 326 |
| PYF Takaroa Atoll 02 | TAKARO02 | -145.0295 | -14.4740 | MG | 4 | False | 2012-11-29 | 2013-09-28 | 303 |
| KIR Abemama 02 | ABEMAM02 | 173.7539 | 0.3922 | TR | 9 | False | 2011-11-01 | 2012-07-07 | 249 |
| PLW Palau 01 | PALAU01 | 134.4944 | 7.3261 | TR | 10 | False | 2012-03-23 | 2012-11-27 | 248 |

| PYF  Raivavae Island 01 | RAIVAV01 | -147.6889 | -23.8825 | MP | 4 | False | 2020-03-11 | 2020-10-04 | 207 |
|---|---|---|---|---|---|---|---|---|---|
| KIR Abemama 01 | ABEMAM01 | 173.8346 | 0.3764 | TR | 9 | False | 2011-11-01 | 2012-04-03 | 154 |
| MHL Majuro 02 | MAJURO02 | 171.0451 | 7.1986 | TR | 20 | False | 2011-05-31 | 2011-10-31 | 152 |
| MHL Majuro 01 | MAJURO01 | 171.0543 | 7.1925 | TR | 4 | False | 2011-05-31 | 2011-10-20 | 142 |
| PYF Vairao 01 | VAIRAO01 | -149.2933 | -17.8064 | MG | 3 | False | 2023-12-28 | 2024-05-16 | 139 |

**Table A5. ReefTEMPS quality flags, derived from NERC Vocabulary Server (NVS) RD2**
**(https://vocab.nerc.ac.uk/collection/RD2/current )**

| Class | Quality | Description |
|---|---|---|
| Flag 0 | No QC done | No quality control has been assigned to this element. |
| Flag 1 | Good data | The element appears to be correct. |
| Flag 2 | « Probably » good data | The element appears to be probably good. Flag 2 data are good data in which some features (probably real) are present but these are unconfirmed. |
| Flag 3 | « Probably » bad data | The element appears doubtful. |
| Flag 4 | Bad data | The element appears erroneous. |


**Appendix B**

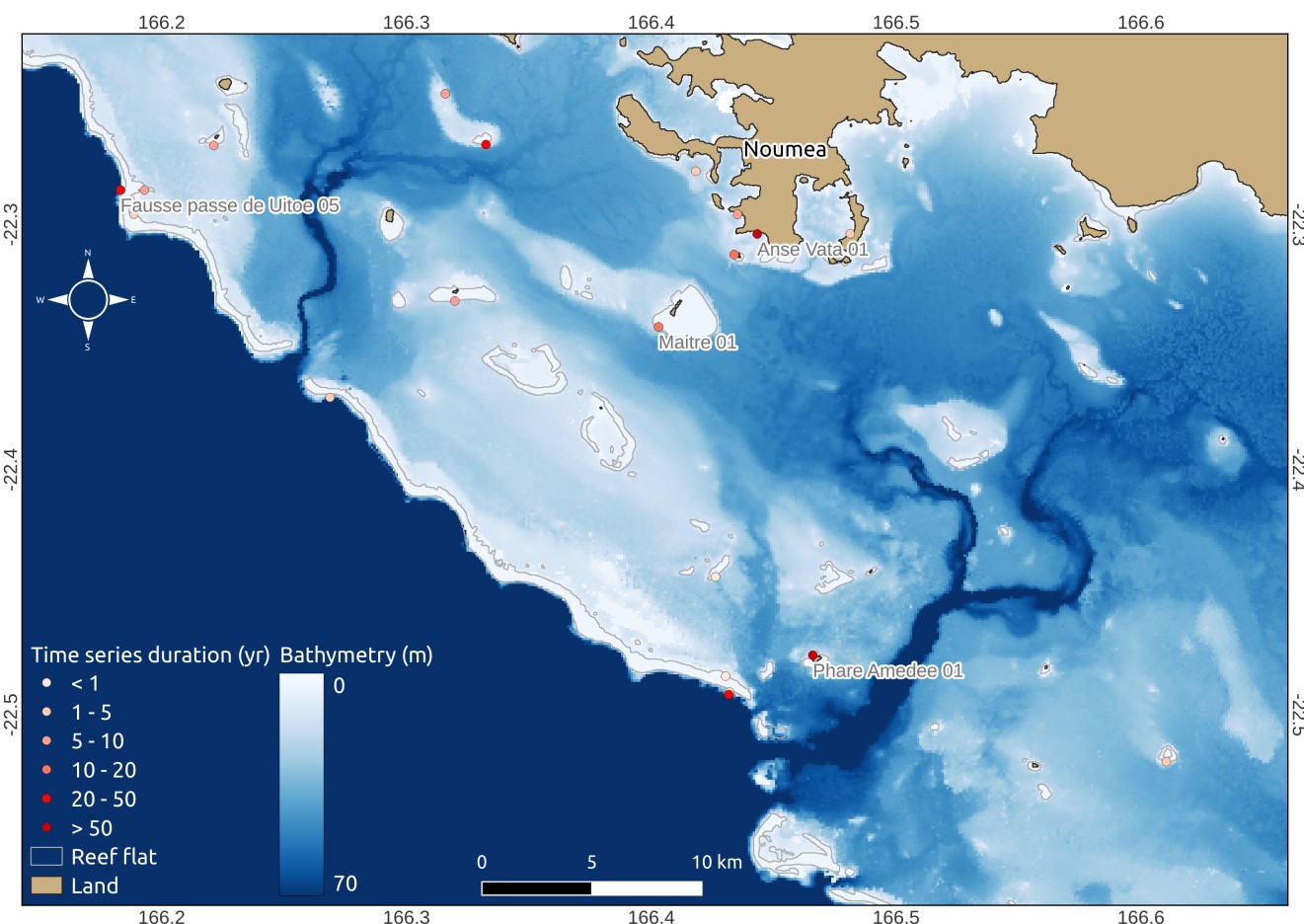

**Figure B1. ReefTEMPS stations in the South-West lagoon of New Caledonia**

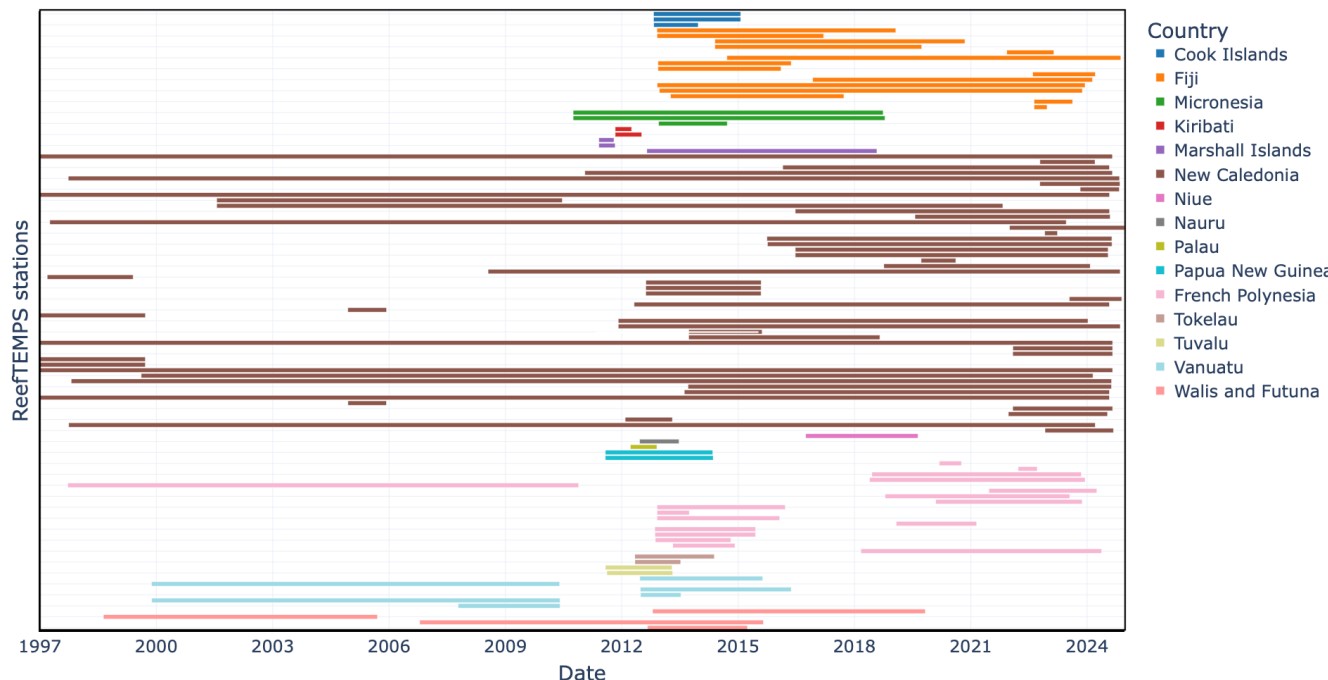

Figure B2. ReefTEMPS Stations activity timeline

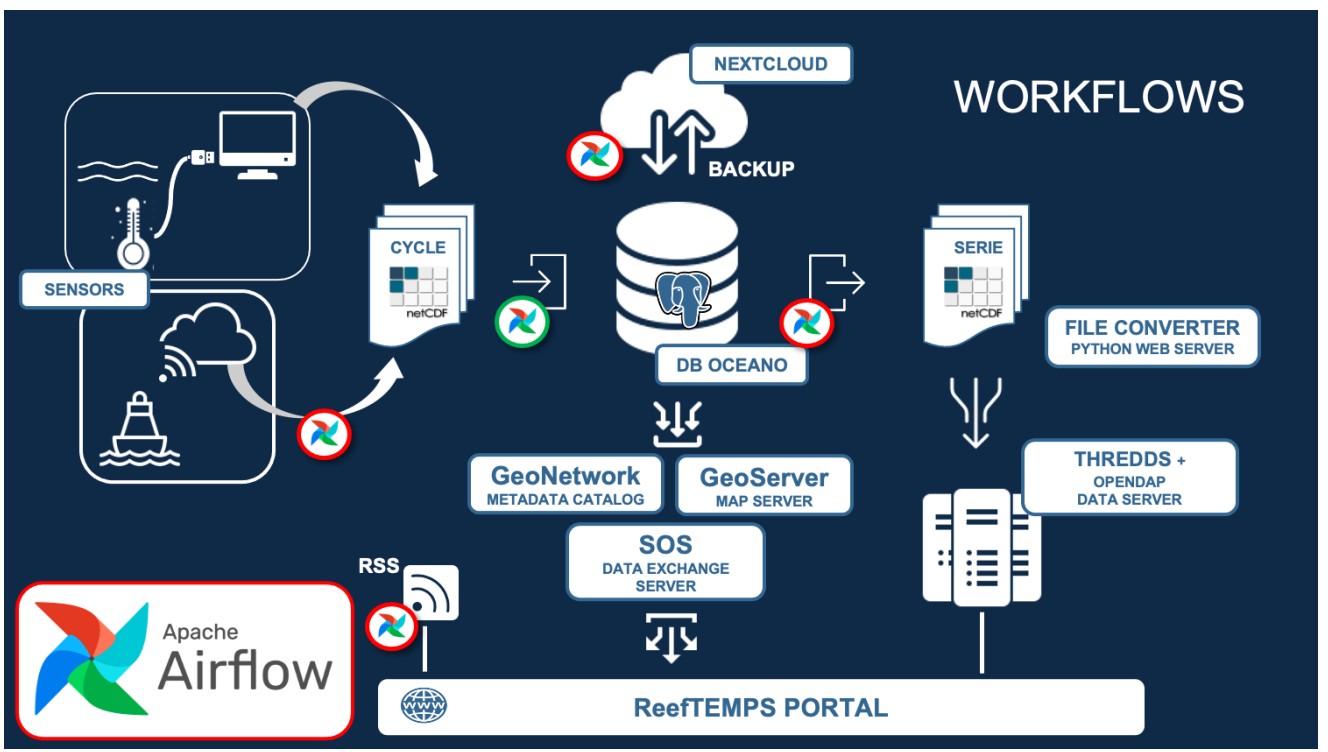

Figure B3. ReefTEMPS data workflow


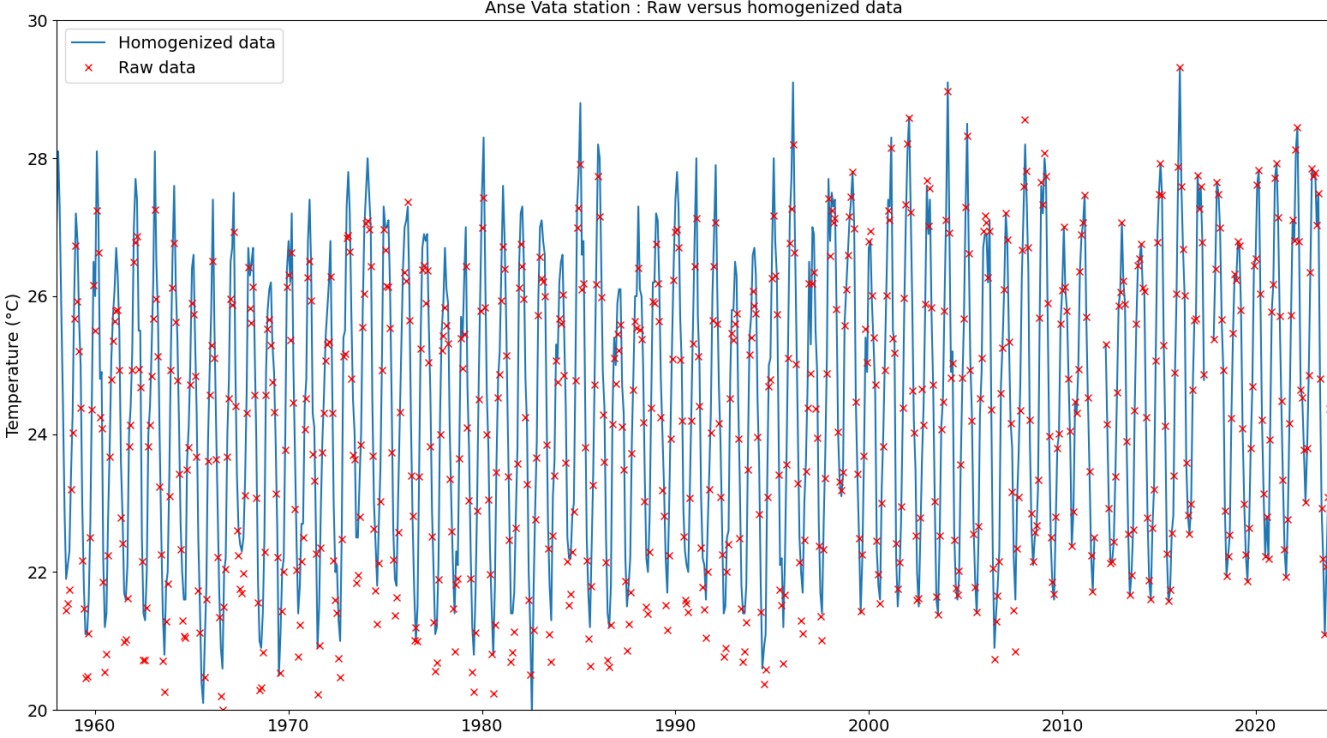

**Figure B4. Raw versus homogenized monthly temperature time series at Anse Vata station.**