# Peer review of "ReefTEMPS: The Pacific Islands Coastal Temperature Network"

_Earth System Science Data, 2024_

## Author Comment (AC1)

**Reply to RC1**: 'Comment on essd-2024-394', Anonymous Referee #1, 27 Nov 2024

We have meticulously addressed each of your comments and suggestions to enhance the quality of our manuscript, the figures and the ReefTEMPS database in general. Please find our replies are shown in red font below.

**Review of the manuscript:**

**ReefTEMPS:** *1 The Pacific Islands Coastal Temperature Network*

*By Romain Le Gendre, David Varillon, Sylvie Fiat, Régis Hocdé, Antoine de Ramon N'Yeurt, Jérôme Aucan, Sophie Cravatte, Maxime Duphil, Alexandre Ganachaud, Baptiste Gaudron, Elodie Kestenare, Vetea Liao, Bernard Pelletier, Alexandre Peltier, Anne-Lou Schaefer, Thomas Trophime, Simon Van Wynsberge, Yves Dandonneau, Michel Allenbach and Christophe Menkes*

This paper presents coastal temperature data from the ReefTEMPS network of moored stations at a number of PICTS in a broad region of the western and central South Pacific, from New Caledonia to French Polynesia. The in situ temperature time series are considered unique in several respects, both in terms of the longevity of some historical stations - the oldest dating back to 1958 and providing daily data for more than 65 years - and in terms of the number of countries surveyed (16 PICTS) and the diversity of coral ecosystems monitored (from atolls to high islands and from outer reef slopes to narrow, shallow lagoons). All data from the stations are publicly accessible via the dedicated ReefTEMPS information system, which also enables rapid visualization of the time series, or via the marine data platform SEANOE. The longevity of these temperature time series makes it possible to diagnose long-term trends, show the influence of various processes on temperature dynamics and document the temporal evolution of extreme events.

**General Comments**

The manuscript (ms) is well structured and written and demonstrates the importance of continuous measurements, especially on moored systems, to determine the physical and biochemical processes that influence biodiversity. However, some improvements should be made before publication; adding some information that I think is missing could improve the text. I found that the authors are vague in some parts of the text and use a very discursive approach without being precise. In particular, in section 3, the authors show the evolution of the sensors used without specifying where (stations, areas) the changes took place. It is difficult to say whether the improvements or changes have taken place at all stations in the network or only in certain areas. See my specific comments.

I appreciate the inclusion of examples of key applications in Chapter 6. Since these are processes determined by long-term measurements on various time scales, it

would be interesting to relate these changes, especially those in temperature, to ENSO. This topic is already briefly discussed on the website, but it would be an added value to discuss it in the handbook.

**Specific comments**

In the abstract:

Line 50: 16 PICTS are indicated, while only 14 are mentioned on the ReefTemps website; which number is correct?

The correct number of PICTS covered by the ReefTEMPS network is 16 (see appendix A1 in supplementary material). The ReefTEMPS portal has been modified accordingly (see https://www.reeftemps.science/). It should be noted that the portal displays 17 countries, as a new monitoring station has recently been added in the Indian Ocean (La Réunion).

Line 57: Quality control is mandatory before sharing data and showing possible trends, while the length of a time series is the added value and the key factor for its determination. Please delete quality control from the sentence.

The reference to "quality control" has been removed from this sentence.

2.1 History:

Figure 1 is missing information mentioned in this chapter as the start of AusAID support in 2011, which led to the deployment of several sensors on different islands. In 2012, the station on Wallis and Futuna was taken over by NC University, so it became the network? In line 57 a dot is missing after GOPS

An arrow indicating the year (2011) when the Pacific network extension began thanks to SPC and AusAid has been added. From our point of view, the network is more a network of stations than of implied organisations. But if we consider the network as being dependent of the number of involved organisations, we can say that ReefTEMPS became a network from 2012 onwards. A dot has been added at the end of this sentence line 57 (and a whitespace removed).

2.2 The current ReefTEMPS network

Line 85: again 16 or 14 PICTS? Please enter the number of actual stations or monitored sites here (active, inactive and dismissed). Later in line 92 you indicate that the New Caledonian components of the backbone component of the ReefTEMPS network include 43 stations, so it would be interesting to indicate the total number of stations belonging to this network right at the beginning of the ms.

The right number of PICTS is definitely 16 (see answer above and Table A1 in supplementary material). The total number of stations has been added in sentence line 85 and the number of actually active stations has been specified in brackets. During the review process some new stations has been implemented. Tables, maps and numbers may slightly change from submitted version.

3.1 Oceanographic bucket

Line 22: Please move "The nominal acquisition time for both stations was 7 am local time and the targeted depth using the bucket was 0.5 m" in line 20 after "...nearly 47 years".

The sentence has been moved, making it easier to read.

3.2 Compact autonomous loggers

Line 27: ".... Were used to monitor coastal temperatures"... please specify where?

The three territories concerned by the use of these sensors have been indicated in the sentence: New Caledonia, French Polynesia, Wallis.

Facilities: It is difficult for the reader to tell at first glance which of the platforms listed in Table 1 are still active. I suggest dividing them into active and inactive or color-coding the two/three cases.

The new table (Table A4 in supplementary material) listing all stations has been reformatted with active stations at the top and also sorted by duration of observations. In addition, a new feature has been added to the data portal page, allowing user to filter active or stopped time series.

4. Processing and quality control

The overall strategy is well described for data measured after 2010; however, I cannot see a clear description of procedures when you recovered the instrument. Have CTD casts been taken each time you recover and redeploy the sensors? How often do you replace the instrument? It seems like you are only removing outliers, what about sensor drift? How do you compensate for any discrepancy (align mismatch) between the end of the time series and the new time series?

CTD casts are not taken each time a sensor is retrieved since this procedure is quite recent. Depending on the location of the stations (territory/remote island), this procedure could not be implemented everywhere. Nevertheless, the ReefTEMPS consortium is doing it utmost to generalize this approach to New Caledonia stations.

The frequency of sensor replacement depends on the type of sensor used. Typically, for today's sensors, the same instrument stays in water approx. for 3 months (JFE Thermosalinograph), 6 months (RBR Temperature/Pressure sensors), up to 1-year (SBE56 Thermistor).

Outliers and drifts are actually not really removed but (if visible by expert check) flagged to "probably" bad data. The ReefTEMPS team is currently considering to deploy homogenization methods on the raw frequency data (which will compensate drift or alignment mismatch) and other statistical tests to further improve the quality of this temperature dataset and move to higher level of processing state. This is part of the planned improvements, but the current temperature quality-flagged dataset is already of a high-quality standard as only a very small number of drifts or misalignments have been observed and flagged.

4.2 The procedure for obtaining the homogenized monthly long-term data is very interesting and impressive. It could be of additional value to show how the data was corrected. I propose to include a time series as an example of a raw and a QC-corrected time series (Anse Vata or Phare Amédée).

A new figure has been added in Appendix C (Figure 2). It allows us to illustrate the differences between the monthly data derived from the raw data (bucket and sensor) and the data after the homogenization procedure. A sentence about this figure has also been added in 4.2.

6.2 Characterize physical processes at various timescales

Line 9-10 please rephrase the sentence and highlight the importance of the parameters time series in identifying physical processes at various time scales.

We are not quite sure we understand this comment, so we could not change the sentence.

Appendix A - Table A1.

Not everyone knows the area in which the stations are located, and it is not immediately obvious from the "nom_station". Please add a first column with the area/region (example Cook Island).

The area/territory concerned by the observations is present in the Station Name column of table A4 and is coded with 3 characters. An additional A1 table has been added with all the territories covered by the ReefTEMPS Network. We thus decided not to include a new column in Table A4.

Add the last sensor type that was used in the table heading after the positions.

See my previous comment on the "active" column

The new A4 table take these comments into account.

Latest type sensor column: explain why you use full names for the sensor types such as "Thermistor" and "Multiparameter" and then abbreviations such as TSG, MG and SEAU (I could not find what this means in Table 3). I suggest standardizing the information in the column and giving the full name for all instruments.

Since the abbreviations are also used for naming the NetCDF files (data portal download or Seanoe), we've added an additional table A2 that describe the instrument types.

End of reply to RC1.

---

## Author Comment (AC2)

**Reply to RC2**: 'Comment on essd-2024-394', Anonymous Referee #2, 17 Dec 2024

We are grateful for the review work carried out by reviewer #2. We have meticulously addressed each of your comments and suggestions to enhance the quality of our manuscript, the figures and the ReefTEMPS database in general. Please find our replies are shown in red font below.

Evaluation of the MS "ReefTEMPS: The Pacific Islands Coastal Temperature Network" by Le Gendre et al.

*General comments:*

The manuscript reports on the ReefTEMPS observation network and focuses on coastal high-frequency seawater temperature time series acquired in 16 Pacific Island Countries and Territories since 1997 and historical daily series back to 1958 for the older. The MS is well written, and embraces a wide scope, which is very much appreciated: from historical context to network and database description, key applications and opening development and perspectives. Section 2 on network description, and section 4 on data processing and quality control are sometimes rather vague or too general to provide a direct and fair vision on current/past sampling effort and data quality.

The data set is impressive and unique by the spatial (mostly intertropical, from roughly 140°E to -178°E) and temporal extent covered (pluri-annual to multi-decadal HF measurements and daily historical time series). Acknowledging the very nice collaborative effort and important work achieved for data aggregation, quality check and dissemination, I also find the dataset of very high relevance. Examples of key scientific applications are well developed (section 6). Such in situ temperature time series are indeed required to accurately characterize local conditions and marine heatwave stress in highly diverse and sensitive coral reef ecosystems, as well as to analyze fine scale coastal processes and dynamics poorly captured by satellite SST. Warming trend analysis carried out at climatic time scale after homogenization of the long-term series is another asset of the MS.

However, some information is missing or lacking precision in the presentation of the network/dataset and regarding the quality check. By adding precise information on total and yearly sampling effort and major breakdown of data series (in particular the proportion still active), authors could greatly ease the understanding and potential reuse of dataset.

The data are available by different means and in different formats (csv, NetCDF), with a dedicated visualization service. Some data series from the NetCDF archive are not shown on the ReefTEMPS portal and the reason why is not clear to me. This should be quantified and explained. Also, the archive comprises 481 nc files in total + one ascii file. For the variable TEMP, there are 185 files with both raw and validated data.

I find the presence of "duplicated" data series with different quality levels in the same directory confusing.

Exploring the dataset itself, 95 files follow the naming convention indicated in section 4.1 (with depth indication), among which 21 raw data series, 57 visually checked (0C) and 17 reduced data sets (3A or 3B). Plotting the visually QC'ed series, I found obviously bad values remaining and flat Quality Flag values at 0. These bad data may fragilize the confidence in the entire data series and impede direct use of the data set, e.g. for satellite data validation. The database needs to be systematically and carefully double-checked and updated by the authors to remove all spurious values and achieve the highest possible number of series visually checked, which to my opinion, and according to Section 4.1. and Figure 3, should be a minimal requirement for dissemination and for publication in ESSD.

We are grateful to reviewer #2 for spending time ensuring quality of the database and pointing out these problems. A great deal of work has been done to remedy these deficiencies (see Introduction part). All the temperature time series has been qualified, and the processing states have been modified accordingly. The introductory paragraph of this response to comments clearly explains the procedure put in place for this. For now, no more "duplicated" series co-exist, all data have been thoroughly checked multiple times and have associated QC flags. This extensive work has significantly improved the quality of the database, which now corresponds much more to the standards required for scientific dissemination and publication.

*Specific comments:*

**Section 2.** The description of the data set is sometimes too vague or general. Complementary information should be provided on the total number of series, equivalent in total year of observation, on the proportion (and quantity) of active vs interrupted data series and major breakdown by origin (sensor type, e.g. tide gauge vs benthic loggers), depth (please consider indicating 0-10 m available as ground-truth for satellite SST), single vs. multiple depth (verticals).

Complementary information has been added in the section 2. More information are now provided about the total number of series, files and stations, the proportion of active and interrupted. The depth categories have been revised to show the proportion of stations between 0-10m, 10-20m and greater than 20m. Information on multiple depth stations has also been added.

Figure 1. Consider showing the number of series available for each year on the timeline.

Appendix A, Table A1. Please consider presenting the table differently, starting with active series and followed by past/interrupted ones. A supplementary summary figure showing data availability by site since 1997 would ideally complement Table A1.

In order not to overload Figure 1, but to provide the information needed to be clarified following the two previous comments, we have added a Gantt style chart in Appendix B1 describing the Sensor Activity Timeline by country since 1997.

Page 6 line 87. Indicate the amount/proportion of stations/series stopped.

The number of interrupted stations has been added.

Page 6 line 93 "and the longest time series" please indicate the number of series with a minimum duration of 10 years

A sentence has been added to specify the number of stations with more than 10 years of data (26 monitoring stations).

Figure 2. Consider using different symbols on the map for ongoing vs. interrupted series (e.g. circles vs. squares).

Figure 2 has been updated to show the status of the stations. Circles represent ongoing time series whereas triangles show interrupted stations.

**Section 4 & 5.** These sections describe the data life cycle, quality control, management and dissemination in a very general way. The total number of data files and major breakdown by QC level should be quantified and explained somewhere, potentially in section 4.1 or in section 5.

The text of section 4.1 has been improved to be more precise and in line with the major update carried out on the database (see introductory paragraph).

Line 73. Indicate the time interval at which maintenance and recalibration were performed, either systematically or in general with some exceptions. If possible, provide feed-back on typical stability and results from intercomparison.

More general information on maintenance has been added. The maintenance/recalibration procedure might evolve throughout the years, devices and countries so it can be difficult to be exhaustive because of so many different cases. Typically, in New Caledonia, an inter-comparison procedure is now applied (since 2021) where sensors are tested against a reference sensor (a brand new SBE56). The tolerated threshold for the deviation from the reference sensor has been set at 0.005°C (see image below).

[Figure]

```
Min difference : −0.00280000000000058

Max difference : 0.004100000000001103

Mean difference : −0.00030818181818188583

Standart Deviation : −0.00030000000000285354

Max. Deviation Tolerance : 0.005

Comparative Test Result :    PASSED
```

**Section 6.1.** Marine heatwaves refer to discrete events with significant deviation to a baseline or climatology. Please consider showing some pluriannual or climatological mean in Figure 5(b,c) and indicate anomaly/intensity in the text in order to figure out how extreme were those events.

Since calculating a robust climatology requires a large number of monitoring years, we were only able to calculate it for the Anse Vata station. It is now displayed in Figure 5.b and the text has been adapted to mention the intensity of the thermal anomaly during the 2016 event in New Caledonia (between 2.5° and 3°C above climatological values during 20 consecutive days).

Page 14 line 66. "Played a key role…". Replace by something more explicit like "negatively impacted the health of ecosystems…".

Sentence has been rephrased using reviewer #2's suggestion.

Page 14 line 90. "large biases (more than 2°C)…". The text on satellite SST vs. in situ comparison is a bit short and should be extended from analysis of Figure 5.

We believe that the aim here is not to be exhaustive on the comparison between in-situ and satellite SST and that the reader can make his own analysis. We decided to keep the text in the original version.

*Technical corrections:*

Page 8 line 21. Repetition of "1977", the sentence should start by "At the Amédée …"

Done.

Figure 5. Consider indicating color legend above panel "a" for New Caledonia, Fiji and French Polynesia, as in Figure 6. Indicate sampling depth directly in panels.

Done.

Figure 6. Indicate depth in panels. The Y-label Temperature (°C) is missing. Consider inserting the legend for the black curves "elevation" and "wind speed" in panels "a" and "c" directly. Satellite SST data in panel "d" are hardly visible.

Figure 6 has been modified accordingly. We agree that it could be difficult to distinguish satellite SST data on this plot but decided to keep the same scale, as the aim of the plot is to show mainly the inter-annual variability observed.

Figure 7. Legend on the trend on yearly warmest months could be removed as it is not showed.

Done.
End of Reply to RC2.

---

## Referee Report (RR1)

2nd Review of the manuscript: ReefTEMPS:

The Pacific Islands Coastal Temperature Network
By Romain Le Gendre, David Varillon, Sylvie Fiat, Régis Hocdé, Antoine de Ramon N'Yeurt, Jérôme Aucan, Sophie Cravatte, Maxime Duphil, Alexandre Ganachaud, Baptiste Gaudron, Elodie Kestenare, Vetea Liao, Bernard Pelletier, Alexandre Peltier, Anne-Lou Schaefer, Thomas Trophime, Simon Van Wynsberge, Yves Dandonneau, Michel Allenbach and Christophe Menkes

The manuscript has significantly improved and most of my comments have been addressed.
In my general comments I wrote:
I appreciate the inclusion of examples of key applications in Chapter 6. Since these are processes determined by long-term measurements on various time scales, it would be interesting to relate these changes, especially those in temperature, to ENSO. This topic is already briefly discussed on the website, but it would be an added value to discuss it in the manuscript.

This point has not been fully addressed in the revised manuscript. While I acknowledge the considerable effort the authors have made to improve the manuscript—and I am generally satisfied with the new version—I still believe that including a section emphasizing the importance of basin-scale measurements would significantly enhance the overall contribution of the work.

**I leave the decision on whether to incorporate this aspect to the discretion of the editor.**

**Minor comments:**

**Line 469:** correct abobe with above

**Line 501–502:**

In my previous comment, I recommended revising the beginning of the paragraph to emphasize the importance of measurements in resolving processes occurring at different temporal scales. However, the current formulation does not sufficiently convey this significance. I suggest revising the start of the paragraph as follows:

*The temperature records from the ReefTEMPS network demonstrate the importance of capturing physical processes operating across multiple temporal scales. These measurements enable the differentiation of high-frequency variability, such as tidal or diurnal fluctuations, from lower-frequency signals associated with seasonal or interannual dynamics, thereby providing a comprehensive understanding of coastal oceanographic processes. Figure 6 shows examples of physical processes affecting temperature at different timescales as captured by the ReefTEMPS network.*

---

## Author Response (AR2)

**Answer to topic editor decision**

We have taken into account the comments made by the topic editor and modified the document accordingly. A sentence on the importance of a basin-wide network has been added. In addition, details of the format and standards used for data dissemination have been added to the Data Availability section.

Best regards

---

## Author Response (AR3)

Dear Topic Editor,

We have taken all your comments into consideration. We apologize for the confusion between Appendix and Supplementary files. The final version of the manuscript does include the tables and figures as Appendices.

Best regards